# RNA structure probing reveals the structural basis of Dicer binding and cleavage

Qing-Jun Luo [1,2,8], Jinsong Zhang[3,4,8], Pan Li [3,4,8], Qing Wang[1,2,5], Yue Zhang[1,2,6], Biswajoy Roy-Chaudhuri[1,2,7], Jianpeng Xu[1,2], Mark A. Kay [1,2,9✉] & Qiangfeng Cliff Zhang [3,4,9✉]

It is known that an RNA's structure determines its biological function, yet current RNA structure probing methods only capture partial structure information. The ability to measure intact (i.e., full length) RNA structures will facilitate investigations of the functions and regulation mechanisms of small RNAs and identify short fragments of functional sites. Here, we present icSHAPE-MaP, an approach combining in vivo selective 2'-hydroxyl acylation and mutational profiling to probe intact RNA structures. We further showcase the RNA structural landscape of substrates bound by human Dicer based on the combination of RNA immunoprecipitation pull-down and icSHAPE-MaP small RNA structural profiling. We discover distinct structural categories of Dicer substrates in correlation to both their binding affinity and cleavage efficiency. And by tertiary structural modeling constrained by icSHAPE-MaP RNA structural data, we find the spatial distance measuring as an influential parameter for Dicer cleavage-site selection.

[1] Department of Pediatrics, Stanford University, Stanford, CA, USA. [2] Department of Genetics, Stanford University, Stanford, CA, USA. [3] Ministry of Education Key Laboratory of Bioinformatics, Beijing Advanced Innovation Center for Structural Biology & Frontier Research Center for Biological Structure, Center for Synthetic and Systems Biology, School of Life Sciences, Tsinghua University, Beijing, China. [4] Tsinghua-Peking Center for Life Sciences, Beijing, China. [5] Present address: Faculty of Preventive Medicine, A Key Laboratory of Guangzhou Environmental Pollution and Risk Assessment, School of Public Health, Sun Yat-Sen University, Guangzhou, China. [6] Present address: Freenome Holdings Inc., South San Francisco, CA, USA. [7] Present address: Impossible Foods Inc., Redwood City, CA, USA. [8] These authors contributed equally: Qing-Jun Luo, Jinsong Zhang, Pan Li. [9] These authors jointly supervised this work: Mark A. Kay, Qiangfeng Cliff Zhang ✉email: markay@stanford.edu; qczhang@tsinghua.edu.cn

Genome-wide RNA structure (the so-called "structurome") studies marry chemical probing with next-generation sequencing[1]. Dimethyl sulfate (DMS), 1-methyl-7-nitroisatoic anhydride (1M7), 2-methylnicotinic acid imidazolide-azide (NAI-N$_3$), and Kethoxal are widely used reagents for RNA structure probing in vivo[2–6]. DMS modifies the N1 and N3 positions of adenine and cytosine bases within single-stranded regions in vivo[2,3], whereas NAI-N$_3$ acrylates the free 2′-hydroxyl groups for all four single-stranded bases, allowing for in vivo structure probing of the transcriptome by selective 2′-hydroxyl acylation followed by Primer Extension (icSHAPE)[4]. icSHAPE has been used to uncover structural variations of RNAs associated with different biological processes, such as translation, RNA-protein interactions, and $N^6$-methyladenosine modification in living cells[7].

Structure probing methods, including DMS-seq and icSHAPE, measure reverse transcription truncations arising at chemically induced nucleotide modifications to determine the probability that a nucleotide is in a single-stranded conformation. A limitation, however, is that structural information at the 3′ terminus of a probing target will be missing, due to the loss of mapping of short sequencing reads (Supplementary Note 1). This target could be an intact transcript or a fragment in a focused study, e.g., a functional region of a long RNA. This caveat becomes a big issue when applying these approaches for structural analysis of targets that are short in size, including small RNAs (sRNAs, RNA of length <~200 nt) or binding sites of RNA-binding proteins (RBPs). To overcome this 3′ terminal drop-off problem, DMS-mutational profiling with sequencing (DMS-MaPseq)[8], and SHAPE-MaP[9] measure the rate of mutations generated during reverse transcription. However, DMS-MaPseq provides partial nucleotide coverage (only adenosine "A" and cytidine "C" nucleotides could be probed) and current SHAPE-MaP reagents (e.g., NMIA and 1M7) have only moderate cell membrane-penetration abilities, limiting their usage in vivo.

Dicer belongs to the RNase III family and cleaves double-stranded RNAs (dsRNAs) and pre-miRNAs into mature small interfering RNAs (siRNAs) or miRNAs, respectively[10–12]. The mature miRNAs/siRNAs load into Argonaute proteins to form a RISC complex, which represses target gene expression directed by Watson–Crick base-pairing between the guide strand and target genes[13]. How Dicer precisely determines its cleavage site on the substrates is of central importance to RNA interference (RNAi) and miRNA biogenesis. Studies have proposed that Dicer measures a certain number of nucleotides from either the 3′ overhang of dsRNA substrates (the 3′ counting rule)[14] or from the phosphate group of the 5′ end for select pre-miRNAs and dsRNAs (the 5′ counting rule)[15]. In addition, our in vivo studies of short hairpin RNAs (shRNAs) and pre-miRNAs revealed that Dicer uses a single-stranded region (a bulge or terminal loop) to precise anchor the cleavage site 2-nt downstream (the loop counting rule)[16]. However, questions remain, in regard to when and to what extent these mechanisms operate in pre-miRNA processing. In addition, Dicer also binds to a variety of substrates without apparent classical miRNA or siRNA processing activities[17], suggesting that it has other roles in RNA metabolism. Whether and how Dicer differentiates between the cleavable and non-cleavable substrates is unknown.

In this work, we present an approach to probe the structures of intact RNAs in living cells. Briefly, we harness the advantages of icSHAPE reagents and of mutational profiling in reverse transcription to develop a structure probing method, icSHAPE-MaP. To demonstrate its capabilities, we use icSHAPE-MaP to determine the complete structural information for cellular sRNAs. In addition, we combined icSHAPE-MaP with RNA immunoprecipitation (RIP) to determine the structural landscape for substrates

of the RNA endonuclease Dicer. By combining structural information obtained by icSHAPE-MaP and tertiary structural modeling, we discover that spatial distance measuring is an influential parameter in Dicer-mediated pre-miRNA processing.

## Results

**Development of icSHAPE-MaP to probe intact RNA structures.** We developed an RNA structure probing method, which we call icSHAPE-Mutational Profiling (icSHAPE-MaP), that uses the icSHAPE reagent NAI-N$_3$ to modify RNA, and subsequently maps mis-incorporation events generated by the reverse transcriptase Superscript II (i.e., reverse transcription mutations) at the nucleotides with NAI-N$_3$-induced RNA modifications (Fig. 1a, see "Methods").

To evaluate the ability of icSHAPE-MaP to capture structural information from intact RNAs, we examined sRNA species (<~200 nt) in HEK293T cells. We performed both in vivo and in vitro icSHAPE-MaP structure probing (see "Methods"). For in vivo probing, NAI-N$_3$ was added directly to cells, which preferentially reacts with free 2′-hydroxyl groups of unstructured and flexible nucleotides, and then the sRNA fraction was purified. For in vitro probing, the sRNA fraction was first purified, refolded, and then treated with NAI-N$_3$ in a tube. The remaining library construction steps were essentially the same for both, where sRNAs were ligated with two adapters at the 5′ and 3′ ends, and reverse transcribed with Superscript II reverse transcriptase. In principle, both reverse transcription mutations (RT-mut) and reverse transcription stops (RT-stop) indicate NAI-N$_3$ modification and hence the structural flexibility of nucleotides. However, Superscript II reverse transcriptase usually adds a random number of non-template nucleotides at the 3′ end of cDNA[18], which confounds accurate RT-stop identification. In icSHAPE-MaP, we thus only used RT-mut for RNA structural probing. To remove RT-stop fragments, we added adapters at both 5′ and 3′ end with polymerase chain reaction (PCR) primers prior to reverse transcription, thus only full-length sequences were amplified for subsequent analysis (Supplementary Fig. 1a).

To map RT-induced mutation sites, we performed deep sequencing and computational analyses, generating an icSHAPE-MaP reactivity/structure score for each nucleotide (see "Methods"). The score negatively correlates with the likelihood of the nucleotide being paired, providing a measure of its secondary structure information. The icSHAPE-MaP experiments were reproducible between independent biological replicates as the mutation rates of each transcript were highly correlated between two replicates (Supplementary Fig. 1b).

We combined the mutational profiles in replicate libraries to calculate structural scores (see "Methods"). We sequenced ~200 M reads and obtained structure scores for 186 transcripts with in vivo samples and 250 transcripts with in vitro samples (Supplementary Fig. 1c, see "Methods"). As an example, we found that icSHAPE-MaP covered almost all (115/121) bases of 5S rRNA (Fig. 1b, c). In contrast, previous work using icSHAPE did not detect about 30 nucleotides at the 3′ end[4]. The scores agreed well with the structure of 5S rRNA, demonstrating the accuracy of icSHAPE-MaP (AUC = 0.825, Fig. 1c, see "Methods"). We also obtained accurate structure scores for other sRNAs with known secondary or tertiary structure models including the 3′ fragment of RNU7 (a small nuclear RNA, snRNA, AUC = 0.994) and Gln-TTG-2-1 (a tRNA, AUC = 0.818) (Supplementary Fig. 1d). Furthermore, when comparing the structure scores to the secondary structure models of tRNAs from GtRNAdb[19], the single-stranded regions had higher icSHAPE-MaP scores than the double-stranded regions ($p$ value < 10e−32, unpaired $t$ test) (Supplementary Fig. 1e).

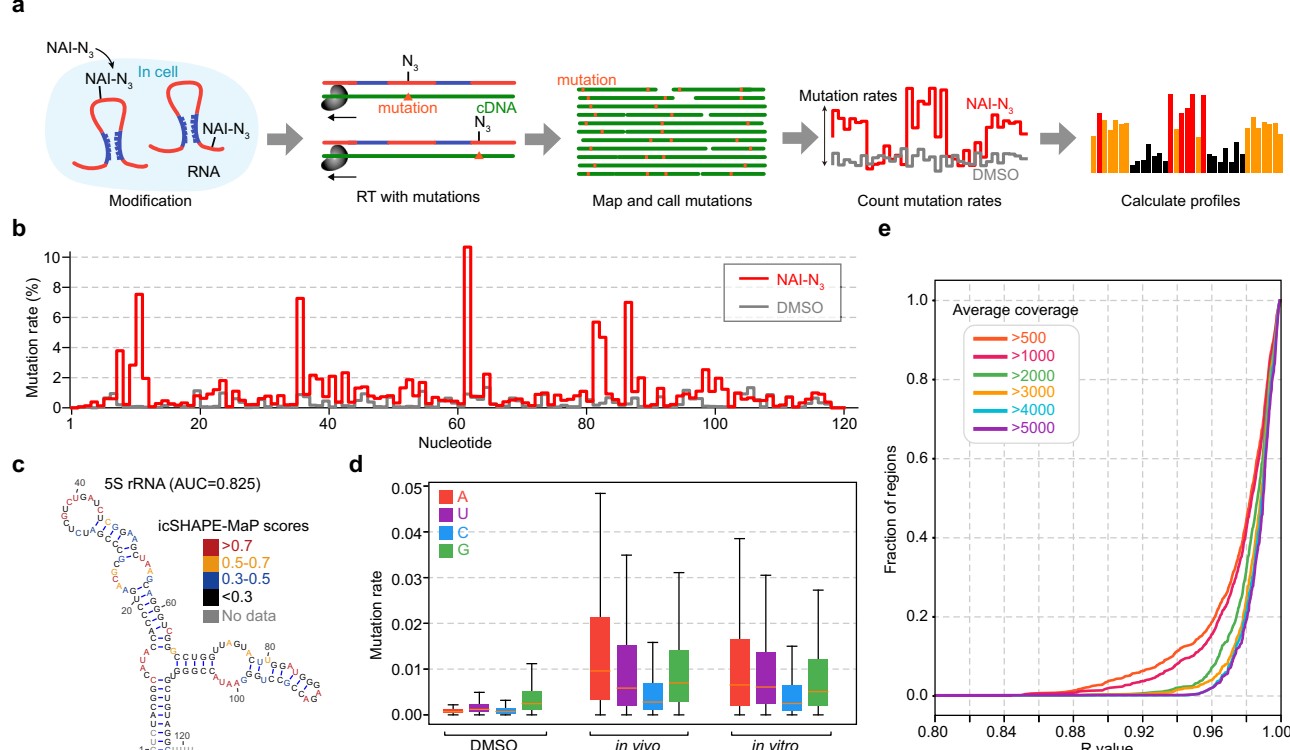

**Fig. 1 icSHAPE-MaP can probe intact RNA structures. a** Flowchart of the icSHAPE-MaP protocol. **b** DMSO (gray) and NAI-N$_3$ (red) mutation profiles of human 5S rRNA. **c** The secondary structure model and icSHAPE-MaP score for each base of 5S rRNA, with different colors representing different range of scores. Gray color indicates a missing score. **d** Box plot of mutation rates of four bases in DMSO ($n = 3156$, 3493, 2978, or 3411, respectively), in vivo ($n = 2871$, 3229, 4002, or 4671, respectively), and in vitro samples ($n = 3117$, 3485, 3049, or 3539, respectively). Box plots are shown with median (orange horizontal line), 25th and 75th percentiles (box edges), and 1.5-fold of the interquartile range (whiskers). **e** The cumulative distribution curve for the Pearson correlation coefficient of NAI-N$_3$ mutation rates between two replicates. Each line represents a different read coverage with different colors. Source data are provided as a Source Data file.

In general, the fraction of mapped reads bearing mutations was substantially higher in the NAI-N$_3$ libraries compared to the control DMSO libraries, both in vivo and in vitro (Supplementary Fig. 1f). Interestingly, some nucleotides in tRNAs displayed high mutational rates in both the DMSO and the NAI-N$_3$ libraries; these sites corresponded to specific endogenous modification sites (Gln-TTG-2-1 is shown as an example in Supplementary Fig. 1g). For example, the 2′-O-methylguanosine (Gm) modification often causes a deletion, with frequent mutations and/or deletions for the 1-methylguanosine (m$^1$G) and 1-methyladenosine (m$^1$A) modifications. This finding underscores the importance of analyzing the background DMSO libraries to determine NAI-N$_3$-independent signals. Our analysis also revealed that the increase in mutational rates was more significant at A and U residues, consistent with previous observations that single-stranded regions are enriched for A/U compared to G/C[4] (Fig. 1d).

NAI-N$_3$ modifications cause various types of mutations, including mismatch, insertion, deletion, and other complex mutations (Supplementary Fig. 1h). We examined the 5S rRNA structure to dissect the different types of mutations and assessed their contributions to the accuracy of icSHAPE-MaP scores. We found that each type of mutation correlated with RNA structure, and in combination they achieved the highest accuracy and coverage in measuring the structure (Supplementary Fig. 1h). We thus used all types of mutations when calculating icSHAPE-MaP scores.

Due to the relatively low mutational rate, a high sequencing depth is needed to obtain accurate structure scores from RT mutational profiling[20]. We estimated the sequencing depth requirement by plotting the cumulative Pearson correlation coefficient ($R$ value) between replicates of regions at different sequencing depths. We found that a cutoff of 2000× sequencing coverage yields very high-quality scores, and that 1000× or even 500× coverage is a reasonable cutoff when considering the trade-off between the cost and the reproducibility (Fig. 1e). Importantly, we found that icSHAPE-MaP required much less sequencing coverage than DMS-MaPseq[20] (Supplementary Fig. 1i).

We also compared the structural differences between in vivo and in vitro icSHAPE-MaP data. We noticed that among all RNA categories, the snoRNAs showed the biggest difference in icSHAPE-MaP scores between in vivo and in vitro conditions (Supplementary Fig. 2), suggesting that their structures vary substantially between these two scenarios. This is consistent with a previously finding that snoRNA structures differ to the greatest extent between in vivo and in vitro conditions compared to other RNA types[4].

**icSHAPE-MaP captures the structural landscape of Dicer binding sites.** We combined icSHAPE-MaP with RIP to profile the RNA structure landscape of Dicer substrates (Fig. 2a, see "Methods"). Briefly, we expressed a catalytic-dead Dicer (D1320A/D1709A) in Dicer-deficient 293T cells and performed RIP to enrich for Dicer-bound transcripts (Supplementary Fig. 3a). Enriched RNAs were then modified by NAI-N$_3$, the sRNA fraction (<~200 nt) was purified and structures were profiled by icSHAPE-MaP.

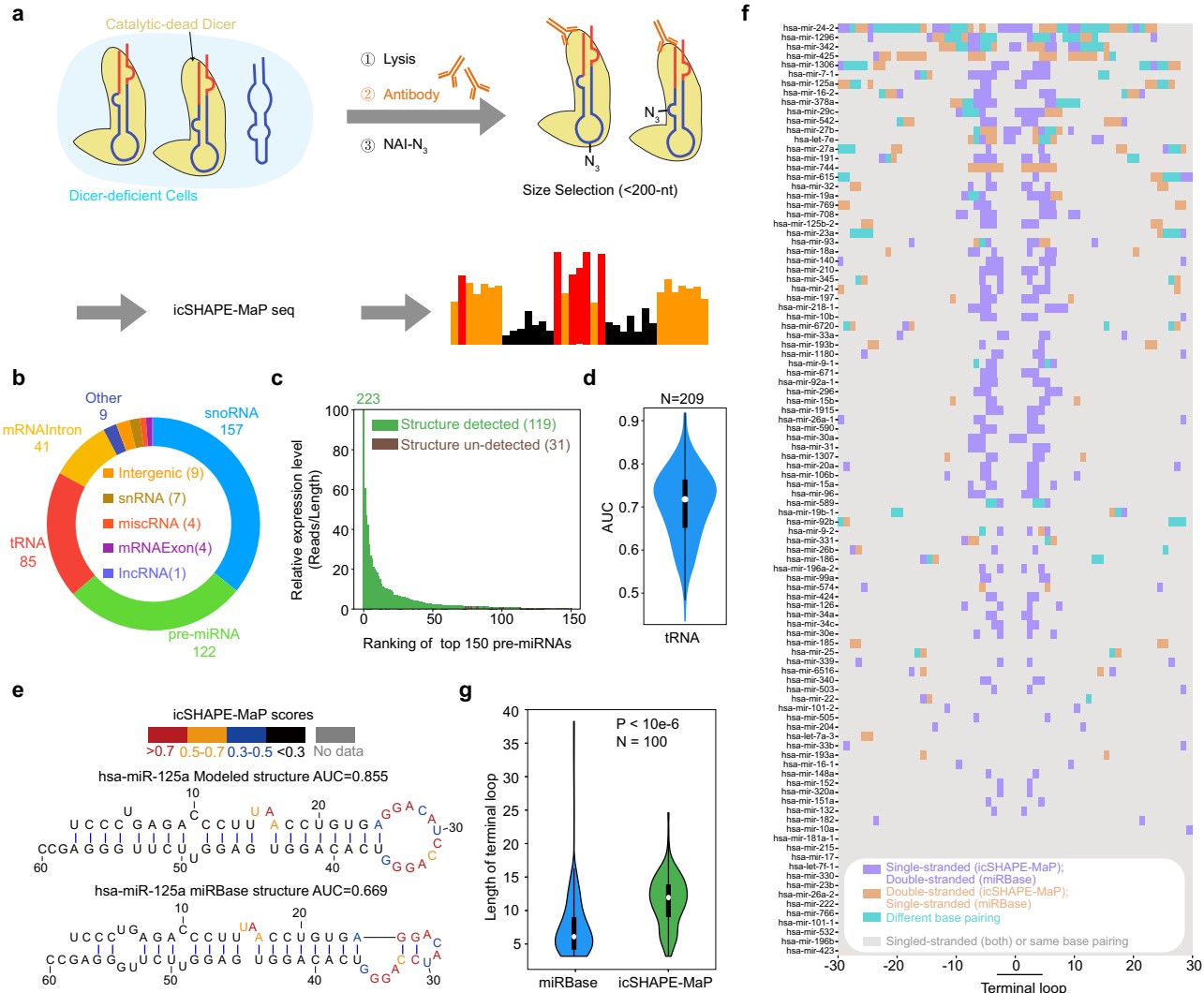

**Fig. 2 RIP-icSHAPE-MaP captures the structural features of Dicer binding sites. a** Flowchart of the RIP-icSHAPE-MaP protocol. **b** Ring diagram of the number of different types of Dicer-enriched RNAs in different colors with icSHAPE-MaP scores. **c** Bar plot of the relative expression level of the top 150 pre-miRNAs, calculated as the number of mapped reads divided by the RNA length. The green bars denote the pre-miRNAs with icSHAPE-MaP scores, and the brown bars indicate those without icSHAPE-MaP scores. **d** Violin plot of Area Under ROC Curves (AUCs) of icSHAPE-MaP scores for 209 tRNA secondary structures from the GtRNAdb database. **e** The secondary structure models and icSHAPE-MaP scores of hsa-miR-125a. The structure on the top was modeled using the RNAStructure software with icSHAPE-MaP scores as constraints; the structure on the bottom is from the miRBase database. **f** A Heatmap highlighting differences between secondary structure models of pre-miRNAs built with or without icSHAPE-MaP scores as constraints. Purple boxes indicate that these nucleotides are single-stranded in the experimentally inferred structures but are base-paired in the structures from the miRBase; orange boxes indicate the opposite. Cyan boxes indicate that these nucleotides are based paired in both structural models, but with different pairing counterparts. Gray boxes indicate those nucleotide positions with identical structures in both models, either single-stranded or base-paired with the same counterparts. **g** A violin plot showing the length distribution of the terminal loop for pre-miRNAs based on the structural models from miRBase prediction (blue) or icSHAPE-MaP (green). *p* Values were determined with a two-sided paired Student's *t* test. Violin plots show the median (white circle), 25/75 percentiles, and the smallest/largest values within 1.5 times the interquartile range above/below quartiles. Source data are provided as a Source Data file.

To define the binding targets of Dicer, we calculated an "enrichment score" as the fold change for target abundance in the "RIP" sample compared to "input" (Supplementary Fig. 3b, see "Method"). We obtained 1,595 enriched RNAs in the unmodified DMSO libraries (Supplementary Fig. 3c). A previous study identified thousands of Dicer binding sites using PAR-CLIP[17]. Despite the different enrichment strategies, the two lists of RNAs agreed well with each other, with more than 50% common premiRNA coverage (Supplementary Data 1). In addition to premiRNAs, we identified other cellular transcripts with a median length of ~70-nt (Supplementary Fig. 3e), including snoRNAs, tRNAs, intronic and exonic sequences of messenger RNAs (mRNAs), and intergenic transcripts. The peak patterns of these

intronic and exonic fragments showed very sharp boundaries with a hairpin-like secondary structure, suggesting that they are functional products processed from their cognate mRNAs (Supplementary Fig. 3d).

Using RIP-icSHAPE-MaP, we obtained the structural information for 820 well-covered (>1000×) RNAs (Supplemental Fig. 3f). We found that the mutational profiles were highly correlated within independent biological replicates, indicating the high reproducibility of our method (Supplementary Fig. 3g, Supplementary Data 2). We used the RIP enrichment scores to classify 439/820 RNAs as Dicer targets, including 122 pre-miRNAs, corresponding to almost every abundant pre-miRNA in 293T cells based on our RNA-Seq data (Fig. 2b, c). As an

internal structure modeling control, we compared the icSHAPE-MaP structure scores of tRNAs in our dataset to their published structure models in GtRNAdb[19] with AUCs calculated (see Methods). Most of the AUCs are well above 0.5 (Fig. 2d), suggesting good agreement between our structure probing and the existing co-evolutionary structure models for tRNAs.

Structure scores from probing experiments can be used to refine RNA secondary structure prediction for physiologically relevant structural models[21]. We used the software suite RNAstructure[22] to predict a local minimum free-energy structure with icSHAPE-MaP scores as a constraint for Dicer substrates. Using this approach, we obtained for example a constrained structural model of pre-miR-125a, which contains a 12-nt terminal loop (G25–G36) (Fig. 2e, top). In contrast, the unconstrained structural model from miRbase (RELEASE 22.1)[23] suggests a smaller terminal loop with multiple bulges and internal loops (Fig. 2e, bottom). In addition, the constrained model of pre-miR-19a shows a 12-nt terminal loop, whereas its miRbase model suggests a small terminal loop, and the constrained model of pre-miR-27b shows a 6-nt terminal loop and a nearby bulge, compared to a small 3-nt terminal loop and one nearby large internal loop in its miRBase model (Supplementary Fig. 3h).

In general, most (78/100) pre-miRNAs have at least one, and almost half (47/100) of pre-miRNAs have at least five structurally different positions between the constrained experimentally inferred and theoretical models. Remarkably, the structurally different positions were often found around the terminal loop region (~±5 nt around the coordinate "0") (Fig. 2f). In general, our experimentally inferred structures modeled a larger terminal loop (median size = 9 nt, Fig. 2g), compared to those from the theoretical model from miRBase (median size = 6 nt). The differences in structural modeling that arises from the pairing of some bases around the terminal loop region by theoretical miRBase models was not supported by icSHAPE-MaP experimental probing. Anecdotally, previous studies suggested that cellular proteins and other factors may help unwind RNA structures so that RNAs are more extended in cells than folded when contained in a test tube[4]. This is consistent with our observations of the more extended terminal loops of pre-miRNAs in our experimentally inferred models compared to the theoretical miRBase models. This analysis highlights the necessity of using experimental information to constrain computational modeling, which will otherwise generate more base-pairings as a result of energy minimization. These results show that the use of icSHAPE-MaP scores can more precisely model RNA secondary structures, providing a structural basis for RNA processing and functional studies.

**Binding preferences and cleavage patterns of Dicer substrates.** To identify the structural determinants of Dicer substrates, we developed a computational method to de novo cluster the icSHAPE-MaP structure profiles (see "Methods"). We first aligned the profiles based on the central loop in predicted structures with constraints of icSHAPE-MaP scores and then performed principle component analysis (PCA) to reduce them into a two-dimensional space. The substrate structures obtained from Dicer RIP experiments established three distinct groups by the K-means clustering, based on the top two principal components (Fig. 3a). Interestingly, the weights of positions in principal component 1 (PC1) of PCA clearly recapitulate the structural feature of pre-miRNAs with a nearly perfect base-paired stem and a relatively large terminal loop (Fig. 3b). Higher icSHAPE-MaP values in the central positions with lower ones in the flanking regions result in more negative PC1 values, which appears to be responsible for the separation of pre-miRNAs vs snoRNAs vs.

tRNAs (Fig. 3a). While in PC2, it shows a chessboard-like pattern with high and low weights occurring in succession, characterizing the cloverleaf structure of tRNAs and separating them from snoRNAs. Accordingly, we found that each cluster was populated with one major type of RNA species: Cluster I was dominated by pre-miRNAs, with Cluster II by snoRNAs and Cluster III by tRNAs.

Cluster I of Dicer substrates showed a relatively large terminal loop with a median size of about 9-nt, flanked by a near-perfect double-stranded stem, which resembles the characteristic hairpin structure of pre-miRNAs (Fig. 3c, d, top panels). In contrast, Cluster II and III consisted of a smaller terminal loop with the median size of 7-nt, surrounded by a loose stem. In addition, a small bulge was observed at around the position +10 for Cluster III, consistent with the cloverleaf structure of tRNAs (Fig. 3c, d, middle and bottom panels). As indicated by the enrichment score, most Cluster I substrates were significantly more enriched than those of Clusters II and III (Fig. 3e, left panel, $p = 2.15e{-}27$ and $p = 1.05e{-}20$, respectively). And substrates from Cluster II are slightly more enriched than those of Cluster III but with no statistical significance (Fig. 3e, $p = 0.21$). Together, these results suggest that perfect hairpin-like structures, e.g., pre-miRNAs, are generally the preferential binding substrates for Dicer.

To directly measure the cleavage activities of Dicer on its substrates, we expressed wild-type (WT) or catalytic-dead Dicer in 293T Dicer-deficient cells, isolated the sRNA fraction of 40–200 nt, and performed RNA-Seq (see "Methods"). As a proxy for the cleavage activities of Dicer, we calculated a "cleavage score" as the log2 fold of change in target abundance in catalytic-dead Dicer vs. WT Dicer cells (Supplementary Fig. 4a, see "Methods"). Increased cleavage scores may suggest the better cleavage activities of Dicer on its substrates. As expected, pre-miRNAs—the major species in Cluster I—showed the highest cleavage activities compared to Cluster II and III (Fig. 3e, middle panel, $p = 2.67e{-}07$ and $p = 1.56e{-}10$, respectively). Likewise, Cluster II had a higher cleavage score than Cluster III (Fig. 3e, $p = 0.01$). These data suggest a direct correlation between Dicer binding affinity and cleavage activity.

We also detected Dicer substrates within mRNA introns and exons that display both high binding and cleavage activities (Supplementary Data 3), including previously reported Dicer substrates such as the 5′ UTR of glutamate-ammonia ligase (GLUL), exon 8 of aurora kinase B (AURKB), and H/ACA box 56 snoRNA (SNORA56)[17], validating our findings. Interestingly, some Dicer cleavage substrates share sequence homology to Alu elements, including intron 4 of the origin recognition complex subunit 5 (ORC5), an intron of the signal recognition particle 68 (SRP68), and an intron of ER lipid raft associated 1 (ERLIN1) (Supplementary Data 3). This may extend the list of Alu units regulated by Dicer, which are implicated in age-related macular degeneration and stem cell proliferation[24,25]. In addition, we detected a Dicer substrate within an intergenic region (chr14:20630387–20630516) that belongs to a repeat family with sequence homology to pre-tRNA-Tyr. These findings suggest additional Dicer functions beyond miRNA biogenesis (also see below).

For further validation of these Dicer substrates and exploration of the regulatory role of Dicer, we performed an in vitro Dicer cleavage assay for selected snoRNAs (Fig. 3f). Interestingly, these snoRNA levels were substantially reduced in the presence of Dicer, similar to our findings with pre-miR-19a (the panel "Full length" in Fig. 3f). Some snoRNAs, e.g., SNORD5, SNORD37, and SNORD56, produced predominant 22-nt sRNAs, with additional minor RNA products of different sizes. To validate this result in cells, we did a qPCR experiment to quantify the expression level of five snoRNAs with Dicer complementation

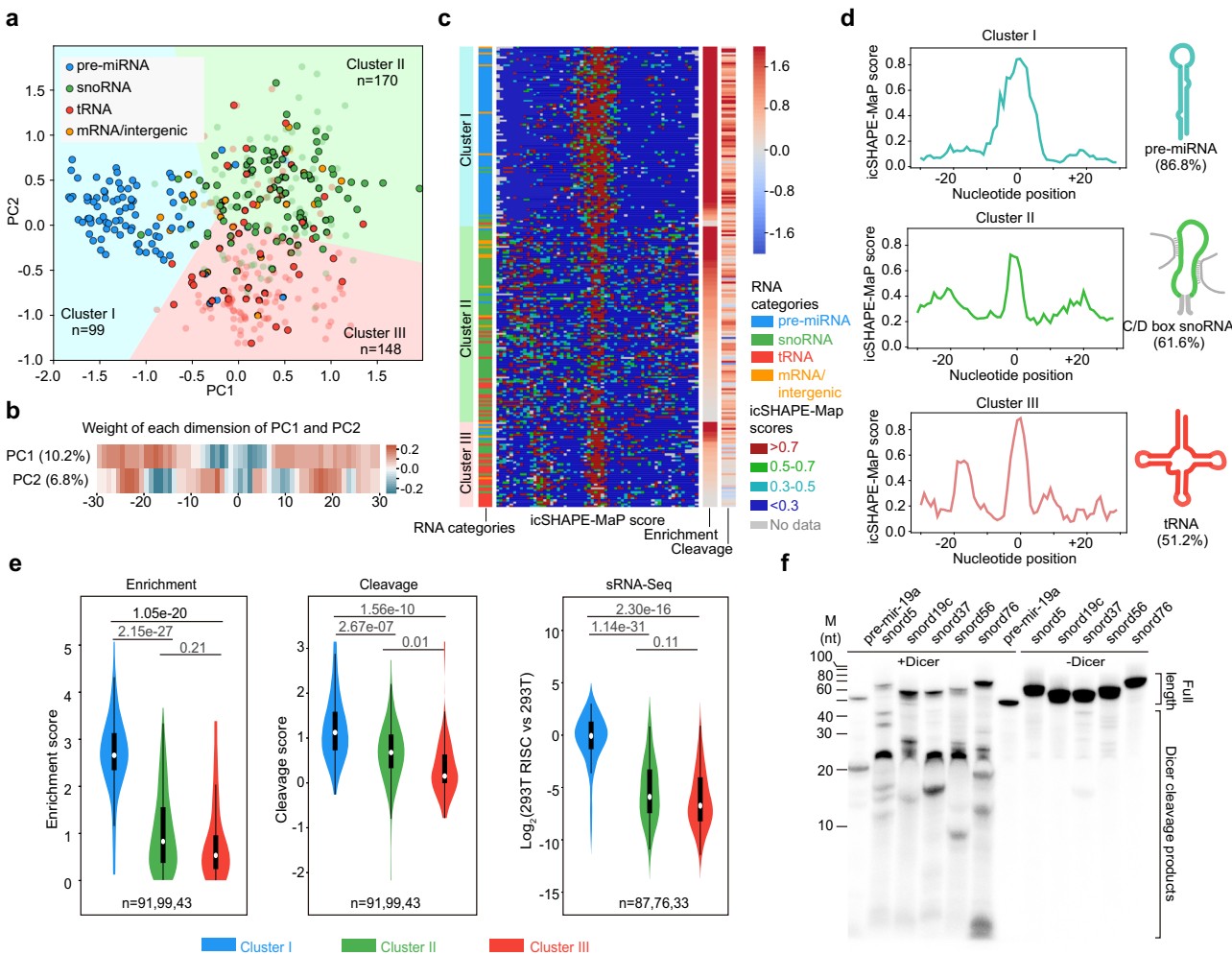

**Fig. 3 Clustering analysis reveals distinct structural features for Dicer substrates. a** Scatter plot of PC1 (10.2%) and PC2 (6.8%) in the PCA analysis using icSHAPE-MaP scores of 30 nt flanking each of the center loops. The dot color indicates its RNA category. Transparent dots suggest that they are not enriched by Dicer RIP. K-means clustering divides all RNAs into three clusters with different background colors. **b** Heatmap of the weights of each position for PC1 and PC2 in the PCA analysis. **c** Heatmap of icSHAPE-MaP scores of Dicer substrates. For each column, the cluster ID, RNA category, icSHAPE-MaP scores, enrichment score, and Dicer cleavage score are indicated. **d** icSHAPE-MaP score profiles (left) and cartoon diagrams (right) for representative RNAs of each cluster are shown. **e** Violin plots of enrichment scores (left, $n = 91$, 99, or 43, respectively), Dicer cleavage scores (middle, $n = 91$, 99, or 43, respectively), and sRNA expression level enriched by AGO2 IP in HEK293T cells (right, $n = 87$, 76, or 33, respectively) for RNAs in the three clusters of Dicer substrates. $p$ values were determined with a two-sided paired Student's $t$ test. Violin plots show the median (white circle), 25/75 percentiles, and the smallest/largest values within 1.5 times the interquartile range above/below quartiles. **f** In vitro dicing assay validated selected snoRNAs as Dicer cleavage substrates. The experiment was repeated two times with similar results. Source data are provided as a Source Data file.

(Supplementary Fig. 4d). In general, their expression was higher in NoDice cells, in which Dicer expression was deficient, compared to 293T cells. In the condition of WT Dicer complementation ("NoDice + WT Dicer"), snoRNA expression levels were greatly reduced, consistent with the data in the in vitro cleavage assay (Fig. 3f). Lastly, the expression levels for these snoRNAs were elevated in 293T cells in which endogenous Dicer expression was knocked-down with siRNAs (Supplementary Fig. 4e). Taken together, these lines of evidence suggest that Dicer is responsible for the stability of these bound snoRNAs.

To determine if these Dicer cleavage substrates are fed into miRNA pathways, we searched publicly available small RNA-Seq datasets of RNAs of ~20-nt[26]. Putative miRNA products matching to Cluster I substrates were enriched in AGO IP samples (Fig. 3e, right panel), lost in Dicer-deficient cells, and recovered their expression when Dicer was rescued (Supplementary Fig. 4b). These data suggest that most of the Cluster I cleavage substrates are *bona fide* precursors for miRNAs. In

contrast, most sRNAs matching to the Cluster II and III substrates displayed an opposite expression pattern, with no or very less enrichment with AGO IP, elevated expression in Dicer-deficiency, and decreased expression when WT Dicer was present (Fig. 3e, right panel, Supplementary Fig. 4b). These included snoRNAs such as SNORA52, SNORD65, SNORD28, and SNORD38B (Supplementary Fig. 4c). Considering their expression level was decreased in our RNA-Seq data for RNAs of ~40–200-nt (Fig. 3c), it suggests that Dicer cleaves these Cluster II and III substrates, independent of miRNA pathways.

**Loop position and physical distance as parameters for Dicer cleavage.** Various models for Dicer cleavage site selection have been proposed, including the 3′ counting rule, the 5′ counting rule, and the loop counting rule[14–16]. However, it remains unclear when and to the extent these rules apply to different substrates. Recently, we solved the structure of Dicer-TRBP in complex with pre-let-7 RNA[27]. It was found that the distance from the 3′ end

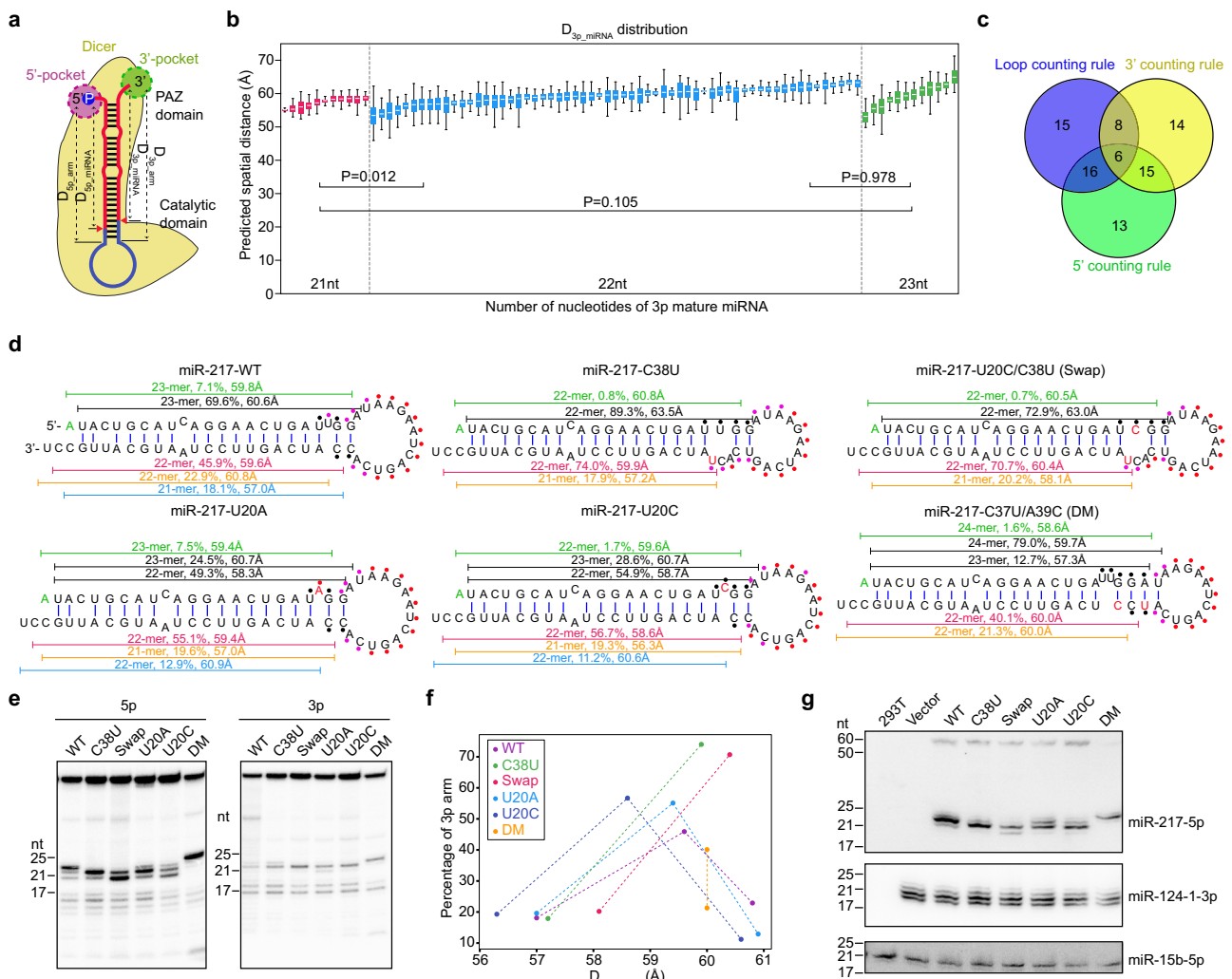

**Fig. 4 Structure modeling reveals critical parameters for Dicer cleavage-site selection. a** Cartoon shows the computational pipeline to calculate the spatial distance for mature miRNAs and their arms of pre-miRNAs. **b** Box plots of spatial distance between Dicer cleavage site on 3p arm of pre-miRNAs and their 3′ end. Each box represents the top 50 3D structures of each pre-miRNA predicted by the Rosetta software. Pre-miRNAs are divided into three groups with the number of annotated nucleotides in length with different colors. *p* Values were determined with a two-sided unpaired Student's *t* test and were adjusted by their false discovery rate (FDR). Boxplots show the median (orange horizontal line), 25th and 75th percentiles (box edges), and 1.5-fold of the interquartile range (whiskers). **c** A Venn diagram shows all logical relations between different rules of Dicer cleavage-site selection on pre-miRNAs obtained by RIP-icSHAPE-MaP. **d** sRNA-Seq for ~20-nt long RNAs indicates a consistency for Dicer cleavage-site selection on 3p arms of pre-miR-217 variants, based on their spatial distance. The relative expression level and distance are shown for each isoform (shown as a blunt-ended line). The nucleotide substitutions on pre-miR217 variants are colored in red. The first nucleotides which may correspond to 5′ end of pre-miR-217 isoforms are colored in green. Reactivity to NAI-N₃, assessed by in vitro SHAPE analysis, are shown as colored dots for select nucleotides. The reactivity is defined as the ratio of darkness quantified by Image J between the NAI-N₃(−) and NAI-N₃(+) sample of the gel. High reactivity (ratio > 1.6), medium reactivity (1.2 < ratio ≤ 1.6) or low reactivity (ratio ≤ 1.2) are denoted as red, orange or black points. **e** In vitro Dicing assay confirms Dicer's direct role on cleavage-site selection shown in panel (**d**). The experiment was repeated three times with similar results. **f** Scatter plot indicating that the isoforms having $D_{3p\_miRNA}$ closer to 59 Å will yield relatively higher expression, compared to their counterparts outside this distance range. **g** sRNA northern blots showing the heterogeneity in the lengths of the miR-217-5p isoforms. mmu-miR-124-1-3p and has-miR-15b-5p were used to show approximately equal loading. The experiment was repeated three times with similar results. Source data are provided as a Source Data file.

binding pocket of Dicer's PAZ domain to the RNase IIIa catalytic site equals that from the 5′ phosphate-binding pocket of the platform domain to the RNase IIIb catalytic site, which is about 58 Å. We hypothesize that Dicer selects the cleavage site by RNA structure, on the basis of a matched physical distance between the pre-miRNA terminus and its processing center, not by a fixed number of nucleotides. We thus set out by characterizing pre-miRNAs based on the loop counting and the physical distance rules, specifically with Dicer cleavage at: (1) 2-nt downstream a bulge or loop (the loop counting rule); (2) a physical distance from the C5′ atom of the 1st nucleotide of miRNA on 3p arm to

the O3′ atom of the 3′ end of pre-miRNA ($D_{3p\_miRNA}$, the 3′ counting rule; Fig. 4a, Supplementary Data 4, and see "Methods"); and (3) a physical distance from the O3′ atom of the last nucleotide of miRNA on 5p arm to the C5′ atom of the 5′ end of pre-miRNA ($D_{5p\_miRNA}$, the 5′ counting rule; Fig. 4a and Supplementary Data 4). With the same methodology, we measured the spatial distance for 5p and 3p arms as well (Supplementary Data 4).

Previously, we observed that ~1/3 3p miRNAs from humans and mice may follow the loop counting rule by using miRBase structure models[16,23]. We revisited this rule using the more

physiologically relevant structural models constrained by the icSHAPE-MaP data (see "Methods"). For each pre-miRNA hairpin, we searched for single-stranded regions on both the 5p and 3p arms. We found that 45% of the 1st nucleotides for 3p miRNAs ("2 nt" in Supplementary Fig. 5a), indicative of Dicer cleavage, are 2-nt downstream of a bulge/loop, following the loop counting rule. In addition to its defined role on 3p miRNA generation, the loop counting rule may apply to 5p miRNAs as well in the form of 0-nt. As a note, 33% of the last nucleotides for 5p miRNAs were found next to a loop/bulge ("0 nt" in Supplementary Fig. 5a), and ~61% pre-miRNAs were shared between these two groups (Supplementary Data 4), showing 2-nt overhangs on the 3′ ends for 5p miRNAs, characteristic features of Dicer cleavage products. However, since we could not rule out the possibility that mis-annotation of the 3′ end of some 5p miRNAs may exist due to post-processing modification(s), we only refer the loop counting rule to its action on 3p miRNAs hereafter. In sum, these data support our previous hypothesis that the loop position of pre-miRNAs is important for Dicer processing.

Next, we examined whether Dicer cleavage-site selection may follow the 5′ and the 3′ counting rules in 3D space. For all pre-miRNAs obtained in our RIP-icSHAPE-MaP experiment, we used the RNAstructure software to construct secondary structural models with icSHAPE-MaP score constraints[22], and used Rosetta to construct tertiary structural models[28] (Fig. 4a and see Methods). We then measured the physical distances between the cleavage sites and the corresponding terminals for 5p and 3p miRNAs from the C5′ atom of the first nucleotide to the O3′ atom of the last nucleotide ($D_{5p\_miRNA}$ and $D_{3p\_rmiRNA}$, see "Methods"). The distances of both $D_{3p\_miRNA}$ and $D_{5p\_miRNA}$ were in proximity to 59 Å for most pre-miRNAs, irrespective of the length of cleaved product (Fig. 4b and Supplementary Data 4). Importantly, for 50/43 of the 100 miRNAs (5p and 3p, respectively), the dicing distance of the miRBase-annotated miRNAs were closer to 59 ± 1 Å (for both 5p and 3p) than the alternative +1 and −1 isoforms (Supplementary Data 4), suggesting a distance of 59 Å as the physical basis for the Dicer cleavage site selection by the 5′ and 3′ counting rules.

We categorized pre-miRNAs based on the rules of Dicer cleavage site selection. As stated above, 45 of them feature the loop counting rule, with 43 following the 3′ counting rule and 50 for the 5′ counting rule (Fig. 4c). A substantial number of pre-miRNAs were shared between each pair of two groups, indicating that different rules collectively exert their influence over Dicer cleavage site selection. We also noticed that 15 pre-miRNAs uniquely follow the loop counting rules (Fig. 4c). Interestingly, most of these produce $D_{3p}\_miRNAs$ that are much shorter than 59 Å (Supplementary Data 4). In other words, a longer isoform is likely closer to 59 Å, but it will result in an RNA containing an additional 1-nt downstream of the terminal loop and violate the loop counting rule (Supplementary Fig. 5b, black dash lines, pre-let-7a-3, pre-miR-204, and pre-miR-101-2 as examples).

**Dicing rules for the processing of pre-miR-217 variants**. To further experimentally evaluate the relationship between different rules of pre-miRNA dicing site selection, we created a series of pre-miR-217 mutants (Fig. 4d), including a C38U mutant that corresponds to a single nucleotide polymorphism (SNP) (dbSNP build 152, rs41291173). We validated the secondary structures of these mutants in vitro by SHAPE analysis (Supplementary Fig. 5c, and see "Methods").

We individually expressed the pre-miR-217 mutants in 293T cells, followed by small RNA-Seq for RNAs of ~20-nt. A constant Dicer cleavage site was detected on the 3p arm (Fig. 4d).

Most of the major isoforms for these mutants had their $D_{3p\_miRNA}$ as approximately 59 Å, consistent with the 3′ counting rule, but do not follow the loop counting rule (see the isoforms colored in red in Fig. 4d). This is further confirmed in the in vitro Dicing assay using chemically synthesized mutants, indicating that Dicer plays a direct role (Fig. 4e, right panel; see "Methods"). This finding suggests that the 3′ counting rule is critical for the 3p arm processing of these variants.

Different potential Drosha cleavage sites may occur for these variants, which confounds the 3′ counting rule. For example, in WT, the 21-mer isoform in blue is only about 57 Å. Extending the isoform to 22-mer is likely closer to 59 Å cleavage distance (Supplementary Fig. 5d, black dash line). However, this 22-mer will reach to the terminal loop. The structural hindrance may affect the tertiary structures so that Dicer chooses the cleavage site producing predominately a 21-mer. This also applies to the mutant of U20A and U20C. Consistent with this notion, only 2% of miRNAs in our RIP-icSHAPE-Map dataset produce their 3p species by the terminal loop based on the miRBase annotation (Supplementary Data 4). In the cases of C38U and Swap, the 3p isoforms with their 5′ ends located at the middle of an asymmetric bulge of 3-nt in size (Supplementary Fig. 5d, black dash lines). This big bulge may also hinder them from Dicer cleavage. In support of this, only two out of ~100 miRNAs (miR-126 and miR-500a) in our RIP-icSHAPE-Map data were produced from a bulge of ≥3-nt in size on their 3p arms, suggesting the processing by Dicer on such a structural feature is rare. Furthermore, the 3p major isoforms for C38U and Swap started with U (Fig. 4d, colored in red and yellow), which may promote loading into AGOs with an extended half-life and subsequent enrichment by small RNA-Seq. Supporting evidence was that the dual luciferase reporter assay showed 3p isoforms with strong biological activity from the C38U and Swap variants (Supplementary Fig. 5e). Finally, the expression of each isoform correlated with its $D_{3p\_miRNA}$ (Fig. 4f). We plotted the percentage of 3p arm reads per isoform divided by total reads per 3p arm as a function of $D_{3p\_miRNA}$. The isoforms with their $D_{3p\_miRNA}$ closer to ~59 ± 1 Å showed a higher level of expression, compared to those not in the range of this distance. Thus, we hypothesize that the 3′ counting rule is the major driving force for 3p isoform production from pre-miR-217 variants.

Interestingly, 5p-miRNA length heterogeneity was observed for different variants of pre-miR-217 (Fig. 4d–g). While U20A, and U20C, WT, and DM-pre-miR-217 generated 22-, 23-, and 24-mers as their major isoforms with the $D_{5p\_miRNA}$ close to 59 Å in length, the 22-mer became the predominant variant for other pre-miRNAs, including C38U and Swap, when the $D_{5p\_miRNA}$ deviated from 59 Å. However, the major 5p isoforms bear a 2-nt overhang at their 3′ ends in accordance with the counterparts of 3p major isoforms, which is a characteristic of dicing by Dicer. This implies that the selection of the Dicer cleavage site for miR-217-5p arms may accompany the selected 3p arm cleavage site.

## Discussion

Our work presents a biotechnology "icSHAPE-MaP" to accurately probe RNA secondary structures in vivo while obtaining the complete information for their intact form. icSHAPE-MaP leverages mutational profiling of reverse transcriptase to detect NAI-N$_3$-induced modifications. Importantly, this method allows for structural analysis of small-sized RNA species, for example full-length sRNAs or fragments (e.g., RBP-binding sites) of longer RNAs. The examples demonstrated in the present study showcase the application of icSHAPE-MaP to unveil the genome-wide structural landscape for Dicer substrate sRNAs. In the future,

icSHAPE-MaP can be applied to reveal the structural features of binding by other RBPs.

In the analysis of the RNA structure of Dicer substrates, we observed that most pre-miRNAs bear a large terminal loop with a nearly perfect stem, structurally and statistically different from other Dicer substrates (Fig. 3b, c). This is reminiscent of the crystal structure of pre-let-7, which displays an A-form helix when loaded into the Dicer-TRBP heterotrimeric complex during the pre-dicing state[27]. Our RIP-icSHAPE-MaP data suggests this structural feature for pre-miRNAs to be a general property of Dicer processing. In terms of Dicer cleavage-site selection on pre-miRNAs, our data suggest that distance counting in three-dimensional space is an important parameter. We hypothesize that Dicer cleaves pre-miRNAs at the phosphodiester bond of the nucleotide with the closest proximity to the physical distance $D_{3p\_miRNA}$ ($59 \pm 1$ Å), upon which mature miRNA lengths are determined (Supplementary Fig. 4f, the middle panel). About 43% of 3p miRNAs observe this rule, counting from their 3′ end (Fig. 4c, the "3′ counting rule")[14], and 50% of 5p miRNAs observe the rule, counting from their 5′ end (the "5′ counting rule")[15]. Together with the "loop counting rule", they coordinately regulate Dicer cleavage-site selection on pre-miRNAs.

Our dicing model is further validated with a series of pre-miR-217 mutants. We found that many pre-miR-217 variants feature the "3′ counting rule". miR-217-3p isoforms are produced at the distance with the closest proximity to ~59 Å, in spite of different nucleotide changes on either 5p or 3p arm, and different Drosha-defined 3′-ends of precursors. Interestingly, miR-217 5p isoforms showed length heterogeneity, with no apparent dominant rules for Dicer cleavage-site selection as a group, implying that Dicer cleavage on 5p/3p arms of pre-miRNAs may not occur simultaneously. Supportive evidence includes that recombinant Dicer generates sRNAs of 24–27 nt from the 5′ arm and 22–23 nt from the 3′ arm of some CNG repeat hairpins with symmetric arms[29]. Furthermore, we observed that pre-miRNAs with long 3p arms tend to display the "3′ counting rule", with a lesser percentage featuring the "3′ loop counting rule" (Supplementary Data 5). The "5′ counting rule" does not correspond to the lengths of either 5′/3′ arm. We hypothesize that these pre-miRNAs would extend part of their arms outside the catalytic domain of Dicer, with the loop loosely interacting with the DExD/H-box helicase domain (Supplementary Fig. 5f, the right panel). In this case, the "3′ counting rule" may play a critical role in the selection of the Dicer cleavage site. For those pre-miRNAs with unusually short arms, their precursors may elbow their way into the processing center of Dicer, whereas the "loop counting rule" and other structural features may dictate the Dicer cleavage site (Supplementary Fig. 5f, the left panel).

Besides its classical role in pre-miRNA processing, Dicer has been reported to produce sRNAs from other cellular RNAs. However, it is still largely unknown whether Dicer possesses endonuclease activities that may not automatically contribute to producing *trans*-acting sRNAs. To address this question, we measured expression levels for Dicer substrates by RNA-Seq with the size selection of 40–200 nt (Fig. 3c and Supplementary Fig. 4a, see Methods). We found that Dicer cleaves snoRNAs, exonic and intronic sequences, and ncRNAs (Fig. 3c, e). In contrast, the concomitant sRNA production from most of these RNAs was not rescued by Dicer reconstitution in Dicer-deficient cells (Supplementary Fig. 4b), with less association with AGO2, based on public datasets (Fig. 3e). In line with this, the expression level of select tRNAs and ncRNAs are increased by RNAi knockdown of Dicer in HEK293 cells and *Caenorhabditis elegans*[17]. In addition, the association with canonical RISC for Dicer-dependent sRNAs from structural RNAs shows species variation. AGOs load the sRNAs from 3′ trailer of isoleucine pre-tRNA and select snoRNAs[30–32], but display poor association with those from

mature tRNA-glutamine[33]. In addition, Dicer-dependent DR2 Alu repeats require AGO3, but not other AGOs, for their precursor maturation and subsequent functions in RNA silencing[25]. Taken together, it suggests that Dicer may dice structural RNAs and other cellular RNAs with hairpin structures and regulate their transcript levels.

## Methods

**Cell culture and transfection**. The 293T cell line was purchased from ATCC (cat. #CRL-3216). The Dicer-deficient 293T cell line (NoDice 2-20) was a gift from Dr. Bryan R. Cullen at Duke University. Cells were maintained in DMEM/HIGH glucose with L-glutamine, sodium pyruvate (Thermo Scientific HyClone), and 10% fetal bovine serum in a humidified incubator at 37 °C with 5% $CO_2$. All transfection assays were done by using polyethylenimine (PEI) (Sigma-Aldrich).

**RNA immunoprecipitation (RIP) and SHAPE modification**. To analyze Dicer substrates at a large scale, NoDice 2–20 cells were transfected with a plasmid expressing human Dicer with two mutations in its RNase III domains (D1320A and D1709A, Addgene). For one 15-cm plate, $9 \times 10^6$ cells were seeded on the first day and transfected with 20 μg plasmids 24 h later with 60 μl (1 μg/μl) PEI. In detail, the plasmids and PEI was first incubated with 1 mL Opti-MEM I Reduced Serum Medium (Gibco) separately. Then the two mixtures were combined and kept at room temperature for 15 min before adding to cells. Forty-eight hours later, cells were lysed in the lysis buffer (50 mM Tris-HCl pH 7.4, 150 mM NaCl, 1% Triton-X 100, 1 mM EDTA), supplemented with the proteinase inhibitor cocktail (Roche) and RNase inhibitor RiboLock (40 U/mL, Thermo Fisher). The lysate was centrifuged at 15,000g for 10 min at 4 °C to remove the insoluble cell debris. The supernatant was incubated with anti-FLAG M2 magnetic beads (Sigma) for 3 h at room temperature.

After incubation, the beads were washed once with the high salt wash buffer (50 mM Tris-HCl pH 7.4, 1 M NaCl, 1% Triton-X 100, proteinase inhibitor cocktail (Roche), RiboLock (Thermo Fisher, 40 U/mL)) and twice with the low salt wash buffer (50 mM Tris-HCl pH 7.4, 150 mM NaCl, 5 mM EDTA, proteinase inhibitor cocktail (Roche), RiboLock (Thermo Fisher, 40 U/mL)). After the last wash, the beads were incubated with the modification buffer (333 mM HEPES, 20 mM $MgCl_2$, 150 mM NaCl, 50 mM NAI-$N_3$) on a Thermomixer at 37 °C for 12 min at 1,000 rpm (NAI-$N_3$ group). For the DMSO group, NAI-$N_3$ was replaced by DMSO in the modification buffer. RNA was isolated with Trizol, following the manufacturer's instruction.

**In vivo and in vitro small RNA modification**. For in vivo small RNA (<200 nt) structure probing, HEK293T cells were treated with NAI-$N_3$ as previously described[4]. Briefly, HEK293T cells were scraped off from culture dishes and washed with phosphate-buffered saline (PBS). Then the cells were resuspended in 100 mM NAI-$N_3$ and incubated at 37 °C on a Thermomixer at 1,000 rpm for 5 min. The reaction was stopped by centrifugation at 2500g for 1 min at 4 °C, and the supernatant was subsequently removed. The cells were resuspended in 250 μl PBS, and then 750 μl TRIzol LS Reagent was added for RNA extraction. The extracted RNA was size-selected on a 6% Urea–PAGE gel for 25–200 nt. For the DMSO control group, the preparation of small RNA was the same as the in vivo modified group except that the HEK293T cells were treated with DMSO instead. For the in vitro group, the RNA extracted from the DMSO group was first heated in metal-free water at 95 °C for 2 min, then was chilled on ice. The 3.3 × SHAPE folding buffer (333 mM HEPES, 333 mM NaCl, 20 mM $MgCl_2$) was added to RNA followed by incubation at 37 °C for 10 min. Then 1 M NAI-$N_3$ was added to a final concentration of 100 mM, followed by incubation at 37 °C for 10 min. Finally, the reaction was stopped by adding 2 × Binding buffer of an RNA concentration kit (Zymo), and the RNA was purified, following the manufacturer's instruction. The details of library construction are described in the following section "Construction of icSHAPE-MaP library".

**Construction of icSHAPE-MaP library**. The RIP pulled-down and "input" RNA was size-selected on a 6% Urea–PAGE gel for 25–200 nt. Gel purification was performed by crushing the gel and incubating in the gel crush buffer (500 mM NaCl, 1 mM EDTA pH 8.0, 10 mM Tris-HCl pH 8.0) with rotation at 4 °C overnight. The eluate was collected by centrifuging through 0.45 μm Spin-X columns (Thermo Fisher), concentrated, and purified by an RNA concentration kit (Zymo). The 8 μL purified RNA was ligated with a 3′ linker by incubation with the 3′ ligation mix (6 μL PEG8000, 1 μL 3′ linker (Supplementary Data 6, 10 μM), 1 μL DTT (100 mM), 2 μL 10 × ligation buffer, 1 μL T4 RNA ligase KQ (NEB), 1 μL RiboLock) at 25 °C for 2 h, followed by inactivation of the enzyme at 65 °C for 20 min. The 1.2 μL RT primer (Supplementary Data 6, 10 μM) was added to the 3′ ligation mix, followed by incubation at 75 °C 5 min, 37 °C 15 min, 25 °C 15 min. The 5′ ligation mix (3 μL PEG8000, 3 μL 10 mM ATP, 1 μL 10 × ligation buffer, 0.5 μL RiboLock, 0.5 μL 5′ linker (Supplementary Data 6, 20 μM), 1 μL T4 RNA ligase I (NEB)) was added directly and incubated at 25 °C for 2 h. The reaction was purified with an RNA concentration kit (Zymo). The 9 μL RT buffer (50 mM Tris-HCl pH 8.0, 500 μM dNTPs, 75 mM KCl, 10 mM DTT, 6 mM MnCl2, 1 μL

RiboLock) was added to the 10 μL purified RNA. The reaction mix was incubated at 42 °C for 2 min. Then 1 μL SuperScript II (Thermo Fisher) was added to the reaction mix. The reverse transcription mix was incubated at 42 °C for 3 h. cDNA products were purified with a DNA concentration kit (Zymo). PCR was set up with 20 μL eluted cDNA and PCR reaction mix (0.5 μL P5 primer (Supplementary Data 6, 20 μM), 0.5 μL P3 index primer (Supplementary Data 6, 20 μM), 0.4 μL 25 × SYBR Green, 20 μL 2 × Phusion HF Master Mix (NEB). PCR was performed in a qPCR machine (Agilent, Mx3000P) to monitor the amplification procedure and was programmed as follows: stage I: 98 °C for 1 min; stage II: 98 °C for 15 s, 65 °C for 30 s, 72 °C for 45 s, with limited cycles. PCR should be stopped before it goes into the log phase. The number of cycles was 13–15. PCR products were purified with a DNA concentration kit (Zymo) and a further size selection (150–330 nt) on a 6% Native PAGE gel. Excess PCR primers were removed and PCR products were purified as described earlier. The library was sequenced on the Illumina HiSeq X TEN platform for paired-end 150 cycles.

**Dicer complementation and RNA-Seq for ~40–200 nt long RNAs**. To study the substrate specificity of Dicer, an RNA-Seq experiment was performed for ~40–200 nt long RNAs as follows. For one well of 6-well plate, 500 ng of plasmids expressing wild type or catalytic-dead Dicer (Addgene, D1320A/D1709A) were transfected into the NoDice 2–20 cells (5 × 10^5). Forty-eight hours later, sRNA (<~200 nt) was extracted using the mirVana kit (Thermo Fisher). One microgram sRNA was then incubated with the end-repairing mix (2 μL 5 × PNK buffer, 2 μL T4 polynucleotide kinase (NEB), 1 μL Fast AP (Thermo Fisher), 1 μL RiboLock, 4 μL nuclease-free water) at 37 °C for 1 h. The 3′ ligation mix (6 μL PEG8000, 1 μL 3′ linker (Supplementary Data 6, preadenylated and blocked at its 3′ end with SpC3, 10 μM), 1 μL DTT (100 mM), 1 μL 10 × RNA ligation buffer, 1 μL T4 RNA ligase I (NEB)) was added directly to the end-repairing mix. The ligation condition was 25 °C for 3 h. The ligation product was size-selected on a 6% Urea–PAGE gel (Thermo Fisher) for ~60–200 nt. Gel purification was performed by crushing the gel and incubating in the RNA elution buffer (20 mM Tris-HCl, pH 7.0; 2 mM EDTA, pH 8.0; 300 mM NaAc, pH 5.2; RiboLock 40 U/mL) with vigorous shaking at 37 °C overnight. The eluate was collected by centrifuging through 0.45 μm Spin-X columns (Thermo Fisher), followed by ethanol precipitation.

For reverse transcription, 9 μL eluted 3′-ligation product was mixed with 1 μL RT primer (Supplementary Data 6, 10 μM). Samples were heated to 70 °C for 5 min in a PCR block, cooled to 25 °C by stepping down 1 °C every 3 s and held at 25 °C. Afterward, 8 μL mutation buffer (50 mM Tris-HCl pH 8.0, 500 μM dNTPs, 75 mM KCl, 10 mM DTT, 6 mM MnCl₂), 1 μL RiboLock, and 1 μL SuperScript II (Thermo Fisher) were added. The reaction mix was incubated at 25 °C for 3 min, and then 42 °C for 3 h. cDNA products were purified by running on 6% Urea–PAGE gel. The following steps, including circularization, PCR amplification and PCR products purification, were done as previously described[4]. Briefly, circularization was conducted as follows, 16 μL cDNA products were mixed with 2 μL 10 × Circligase buffer, 1 μL Circligase II, 1 μL 50 mM MnCl₂ and incubated at 60 °C for 2 hr. The circularization reaction was purified with a DNA concentration kit (Zymo). PCR was set up with 20 μL eluted circularized cDNA and PCR reaction mix (0.5 μL P5 primer (Supplementary Data 6, 20 μM), 0.5 μL P3 index primer (Supplementary Data 6, 20 μM), 0.4 μL 25 × SYBR Green, 20 μL 2 × Phusion HF Master Mix (NEB). PCR was performed in a qPCR machine (Agilent, Mx3000P) to monitor the amplification procedure and was programmed as follows: stage I: 98 °C for 1 min; stage II: 98 °C for 15 s, 65 °C for 30 s, 72 °C for 45 s, with limited cycles. PCR should be stopped before it goes into the log phase. The number of cycles was 13~15. PCR products were purified with a DNA concentration kit (Zymo) and a further size selection (150–330 nt) on a 6% Native PAGE gel. Sequencing was performed on an Illumina NextSeq 500 machine (1 × 75-bp) at the Stanford Functional Genomics Facility.

**Plasmid vector construction**. Cloning of luciferase reporters for miRNA binding sites was done as previously described[16]. Briefly, to express different pre-miRNAs, a mammalian expression vector pBudCE4.1 was used (Thermo Fisher). PCR primers were designed to amply the genomic regions flanking pre-miRNAs (~200-nt upstream and downstream). Mouse *mir124-1* was cloned downstream of the CMV promoter between *Hin*d III and *Bam*H I to serve as a control for transfection and expression experiments. Pre-miR-217-WT was cloned downstream of the EF-1α promoter between *Not* I and *Kpn* I. All other variants of pre-miR-217 were made using the QuikChange Lightning Multi Site-Directed Mutagenesis Kit (Agilent). All oligo sequences for cloning can be found in Supplementary Data 6. The dual-luciferase reporter assay was performed according to our published protocol[16], with the exception that luciferase activities were measured 48 h after cotransfection of 100 ng pre-miRNA plasmids and 100 ng corresponding reporter plasmids.

**Small RNA-Seq for ~20 nt long RNAs**. Library construction, Illumina sequencing and data analysis for small RNA-Seq were conducted as previously described[16]. Briefly, total RNA was extracted with the mirVana miRNA Isolation Kit (Thermo Fisher). The 3′ linker ligation was set up in 20 μL for 16 °C overnight, using 5 μg total RNA and the ligation mix (3 μL PEG8000, 0.5 μL Universal miRNA Cloning Linker (NEB), 100 mM DTT, 1 × RNA ligation buffer, 1 μL T4 RNA Ligase 2 with truncated KQ (NEB), 0.4 U/μL RiboLock)). The ligation product

was size-selected on a 15% Urea–PAGE gel (Thermo Fisher) for ~17–35 nt long sRNAs. Gel purification was performed by crushing the gel and incubating in the RNA elution buffer (20 mM Tris-HCl, pH 7.0; 2 mM EDTA, pH 8.0; 300 mM NaAc, pH 5.2; RiboLock 40 U/mL) with vigorous shaking at 37 °C overnight. The eluate was collected by centrifuging through 0.45 μm Spin-X columns (Thermo Fisher), followed by ethanol precipitation. The 5′ adapter ligation was set up in 20 μL for 16 °C overnight, using the 3′ ligation products and the ligation mix (3 μL PEG8000, 1.25 μL 5′ adapter (Supplementary Data 6), 1 × RNA ligation buffer, 1 mM ATP, 1 μL T4 RNA Ligase 1 (NEB), 0.4 U/μL RiboLock (Thermo Fisher)). The ligation products were subject to ethanol precipitation. Reverse transcription (RT) was set up with RNAs ligated with adapters, 1 μL RT primer, 1 × SSIV buffer, 500 μM dNTP, 5 mM DTT, 1 U/μL RiboLock, and 1 μL SuperScript IV (Thermo Fisher) in a total volume of 20 μL. The reaction was programmed as follows: 25 °C for 3 min, 42 °C for 10 min, 50 °C for 30 min, and 80 °C for 10 min. RT products were then ethanol precipitated and dissolved in 10 μL H2O. To set up PCR reactions, 5 μL of RT products were mixed with 1 × Phusion High-Fidelity PCR Master Mix with HF Buffer (NEB), 200 μM Solexa Forward and Reverse primers in a total volume of 50 μL. PCR was programmed as follows: stage I: 98 °C for 30 s; stage II: 98 °C for 10 s, 52 °C for 30 s, 72 °C for 15 s, repeating for total 22 cycles; Stage III: 72 °C for 5 min. The PCR products were size-selected on a 4% NuSieve GTG Agarose gel (Lonza) for ~125–150 nt and purified with the MinElute gel extraction kit (Qiagen). Multiplex sequencing was performed on an Illumina NextSeq 500 machine (1 × 75-bp) or on a MiSeq machine (1 × 36-bp) at the Stanford Functional Genomics Facility.

**Immunoprecipitation and in vitro Dicer processing**. Immunoprecipitation of Flag-Dicer and in vitro processing for pre-miRNAs were conducted as previously described[15], except for the following modification. pCAGGS-FLAG-hsDicer (Addgene #41584) was used to transfect HEK293T cells. To reduce the viscosity, cell lysate was subject to sonication on ice (Qsonica, amplification 80%, 45 s). The supernatant was incubated with 40 μL of anti-Flag M2 affinity gel (Sigma) with constant rotation for 2 h at 4 °C. The reactions were performed in a total volume of 30 μL in 2 mM MgCl₂, 1 mM DTT, 5′/3′-end-labeled pre-miRNA of 1 × 10^4 to 1 × 10^5 c.p.m. and 15 μL of the immunopurified Dicer in the reaction buffer (200 mM KCl; 20 mM Tris-HCl, pH8.0; 0.2 mM EDTA, pH 8.0). The reaction mixture was incubated at 37 °C for 60 min, followed by the addition of 2 × loading buffer (Thermo Fisher) and separation on 18% polyacrylamide gel with 8M urea.

**Immunoblotting**. Immunoblotting was used to verify the enrichment of FLAG-tagged Dicer in RIP. The experiment was performed with primary antibodies for GAPDH (Abcam, cat. #ab181602, diluted 1:10,000), DICER (Proteintech, cat. #20567-1-AP, diluted 1: 500), FLAG tag (Sigma-Aldrich, cat. #F1804, diluted 1:1,000), and secondary antibodies: goat anti-mouse IgG (H + L)-HRP Conjugate (EASYBIO cat. #BE0102, diluted 1:2,000); Goat anti-rabbit IgG (H + L)-HRP conjugate (EASYBIO cat. #BE0101, diluted 1:2,000). 1% of the sample volume of input or pull-down samples was used for immunoblotting.

**Small RNA northern blots**. 293T cells in 6-well plates were transfected with 2 μg of pre-miRNA-expressing plasmids/well. Total RNA was isolated 48 h after transfection by using the mirVana miRNA Isolation Kit (Thermo Fisher). Totally, 10–30 μg RNA was run on an 18% (w/v) polyacrylamide/8 M urea gel for electrophoresis, transferred onto an Amersham Hybond-XL membrane (GE Healthcare), and blotted with P^32-labeled DNA probes (see Supplementary Data 6 for sequences).

**In vitro SHAPE analysis**. RNA structure analysis was performed by selective 2′-hydroxyl acylation analyzed by primer extension (SHAPE). 1 pmol of RNA was modified with 1 μL NAI-N₃ (5%) or treated with 1 μL DMSO (unmodified control) in presence of 3 × SHAPE buffer (333 mM HEPES, pH 8.0; 20 mM MgCl₂, 333 mM NaCl) for 15 min at 37 °C. RNA was extracted in phenol:chloroform and resuspended in 5 μL RNase-free water or 4 μL for sequencing lanes. A DNA primer complementary to the Universal miRNA Cloning Linker (NEB) ligated to the 3′ end of RNA was 5′ labeled with T4 polynucleotide kinase (NEB) and γ-P^32-ATP. 1 μL of the 5′ labeled primer (~250 nM) was annealed to the resuspended RNA samples at 95 °C and slowly cooled down to ~40 °C. Finally, primer extension was carried out in presence of Superscript III (Thermo Fisher), first-strand buffer, DTT, RiboLock, and 0.5 μL of 2 mM dNTP (plus 1 μL of 10 mM ddNTPs for sequencing lanes) at 55 °C for 15 min in a 10 μL final volume reaction. The reaction was terminated with 1 μL of 4 M NaOH and incubated at 95 °C for 5 min, followed by ethanol precipitation on dry ice. DNA pellet was resuspended in 10 μL of 2 × Gel Loading Buffer II (Thermo Fisher) and separated on 15% Urea–PAGE gel. All (−) lanes were those from DMSO-treated samples.

**qPCR for small RNAs**. Cells were harvested and total RNA was isolated using TRIzol (Thermo Fisher). Isolation of small RNAs from total RNA Samples was done with the *mir*Vana miRNA Isolation Kit (Thermo Fisher). Complementary DNA (cDNA) synthesis was performed after DNase treatment (TURBO DNA-free Kit, Thermo Fisher) using Superscript IV with the random oligo. qPCR analysis

was performed with iTaq Universal SYBR Green Supermix and Bio-Rad CFX384 cycler as described in the manufacturers' instructions.

**siRNAs knockdown against Dicer**. HEK293T cells were transfected with DICER1 (Dharmacon, catalog number: LQ-003483-00-0002) or control siRNA (Dharmacon, catalog number: D-001810-01-05, 80 nM final concentration) by Lipofectamine 3000 (Thermo Fisher). Totally, 72 h after transfection, RNA extraction and RT-qPCR were conducted as described above.

**RNA-Seq analysis**. The sequencing data were processed by removing adapters with Cutadapt (Version 1.16)[34], filtering high-quality reads with Trimmomatic (Version 0.33)[35], and duplication removal with an in-house Perl script (available upon request). Clean reads were mapped to the reference sequences (described in the part "icSHAPE-MaP score calculation") with STAR[36] by default settings. The read count was calculated using an in-house Perl script (available upon request). The enrich score and cleavage score calculation is described in the part "Calculation of the RIP enrichment scores and cleavage scores for Dicer". The transcripts with enrichment scores > 0 were defined as RIP-enriched ones. sRNA-Seq data analysis was done by following our published protocol, except that only reads with the perfect match to pre-miRNAs were used[16].

**icSHAPE-MaP score calculation**.

1. Pre-process. The sequencing data was processed by removing adapters with Cutadapt (v1.16)[34], filtering high-quality reads with Trimmomatic (v0.33)[35], and duplication removal with an in-house Perl script (available upon request).
2. Mapping. Human sRNAs sequences less than ~200-nt in length were collected, such as miRNAs (from miRbase v22)[23], snoRNAs (from Gencode v26)[37], snRNAs (from Gencode v26), tRNAs (from GtRNAdb v2.0)[19], vault RNAs (from RefSeq v109)[38], Y RNAs (from RefSeq v109), and 5S rRNAs. The processed reads as above were mapped to them with STAR (v2.7.1a)[36] with parameters --outFilterMismatchNmax 3 --outFilterMultimapNmax 10 --alignEndsType Local --scoreGap -1000 --outSAMmultNmax 1. To find out other sRNA fragments with limited annotation on the human genome, the unmapped reads were mapped to human genome (version GRCh38.p12) for the reiteration of data analysis mentioned above.
3. Calculate icSHAPE-MaP scores. Replicate samples were combined (with samtools merge)[39]. Shapemapper2 (v2.1.4)[40] was used to calculate final scores as follows:
   a. Mutations on each read were parsed with shapemapper_mutation_parser. The script counts 8 mutation types: mismatch, insertion, deletion, multi-mismatch, multi-insertion, multi-deletion, complex-insertion, and complex deletion.
   b. Number of mutations for each nucleotide was counted with shapemapper_mutation_counter.
   c. icSHAPE-MaP reactivity scores were calculated with make_reactivity_profiles.py.
   d. Raw scores were normalized with normalize_profiles.py.

The calculation process for each base can be briefly summarized by the following formula:

$$s_i = \frac{r\_nai_i - r\_dmso_i}{f} \qquad (1)$$

The icSHAPE-MaP score for base $i$ is the difference between mutation rate in NAI sample and DMSO sample for base $i$ divided by the normalization factor $f$.

**Correlation of mutation rates between replicates**. The total read counts from two replicates were balanced by down-sampling. All bases were sorted by coverage. The bases with coverage greater than 500, 1000, 2000, 3000, 4000, or 5000 were selected to calculate the replicate correlation of mutation rate with sliding window (window size: 50 nt; window step: 10 nt). Finally, the data under each cutoff was used to generate a cumulative distribution curve.

**Computational prediction of RNA secondary structures with constraints**. The Fold program in the RNAstructure package (v5.6)[22] was used to predict the secondary structures of RNAs. The icSHAPE-Map scores were used as constraints with parameters: -si -0.6 -sm 1.8 -SHAPE icSHAPE-Map.shape -mfe.

**RNA secondary structure visualization**. RNA secondary structures were visualized with VARNAv3-93[41] command line. Colors of bases were applied with parameter "-basesStyle1 on and -applyBasesStyle1on".

**PCA analysis and structure clustering**. The secondary structures of RNAs were predicted with icSHAPE-MaP scores as constraints. They were aligned to the center of their central loop and the icSHAPE-MaP scores of flanking 30-nt were used for PCA analysis in the Sklearn (v0.20.3) package. The raw 60-dimensional

space was reduced to a 2-dimentional space. K-means clustering (K = 3) was carried out with 2-dimentional vector in the Sklearn package.

**Calculation of the RIP enrichment scores and cleavage scores for Dicer**. Reads were pre-processed as described above, followed by mapping to reference sequences with STAR. The RIP enrichment score for RNA $i$ was calculated as

$$\text{Enrich score}_i = \log_2\left(\frac{\text{RIP}_i/\text{RIP}_{\text{total}}}{\text{DMSO}_i/\text{DMSO}_{\text{total}}}\right) \qquad (2)$$

The number of reads mapped to RNA $i$ was denoted as $\text{RIP}_i$ in the Dicer RIP sample. The total number of mapped reads in the Dicer RIP sample was denoted as $\text{RIP}_{\text{total}}$. Similarly, the Dicer cleavage score was calculated as

$$\text{Cleavage score}_i = \log_2\left(\frac{\text{Dead}_i/\text{Dead}_{\text{total}}}{WT_i/WT_{\text{total}}}\right) \qquad (3)$$

**Computational prediction of pre-miRNA tertiary structures**. de novo tertiary structure models of pre-miRNAs were produced by Rosetta[42] from secondary structures with constraints from icSHAPE-MaP data. First, RNA helices were built using a custom Python script adopted from Rosetta (available upon request). These helices were fixed throughout the following modeling steps. Second, de novo modeling was done through an established algorithm, Fragment Assembly of RNA with Full Atom Refinement (FARFAR)[42]. A Python script (FARFAR_setup.py) was used to call the Rosetta-built pipeline for FARFAR. 500 tertiary structures for each pre-miRNA per modeling run were produced. The top 50 structures with the lowest Rosetta energy were used for the distance measurement of the Dicer cleavage site (see below).

**Calculation of distance between 5′/3′ ends of pre-miRNA hairpin to the Dicer cleavage site**. The Dicer cleavage sites on pre-miRNAs were inferred from miRBase, based upon the annotated 3′ end of 5p or the 5″ end of 3p miRNA sequences. The distance of Dicer cleavage was calculated using the physical distance between the fifth carbon atom (C5′) on the first nucleotide and the third oxygen atom (O3′) on the last nucleotide for each miRNA. The median distance was presented from the measurement of top 50 predicted tertiary structures for each miRNA. To calculate the arm length, it was measured for the physical distance between C5′ on the first nucleotide and O3′ on the last nucleotide next to the terminal loop on 5p arm ($D_{5p\_arm}$) or for that between C5′ on the first nucleotide next to the terminal loop on 3p arm and O3′ on the last nucleotide of each pre-miRNA ($D3_{p\_arm}$).

**Reporting summary**. Further information on experimental design is available in the Nature Research Reporting Summary linked to this paper.

## Data availability
We have used publicly available datasets of RNA modifications[43] (http://genesilico.pl/modomics/), miRNA sequences, and secondary structure models[23] (http://www.mirbase.org/), and tRNA sequences and secondary structure models[19] (http://gtrnadb.ucsc.edu/), snoRNA and snRNA sequences[37] (https://www.gencodegenes.org/) and vaultRNA and YRNA sequences[38] (https://www.ncbi.nlm.nih.gov/refseq/). The source data for Figs. 1–4, as well as Supplementary Figs. 2–5, is provided as a Source Data file. icSHAPE-MaP and RNA-Seq data that support the findings of this study have been deposited in GEO with the primary accession codes GSE146952. The data supporting the findings of this study are available from the corresponding authors upon reasonable request.

## Code availability
The source code for the analysis is available on Github (https://github.com/lipan6461188/RIP-icSHAPE-MaP) or Zenodo (https://doi.org/10.5281/zenodo.4680657)[44].

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

## Acknowledgements

We thank B. Cullen for the Dicer-deficient 293T cell (NoDice 2–20), and Hongwei Wang for helpful discussions. This work was supported by the National Natural Science Foundation of China (Grant Nos. 91740204, 91940306, and 31761163007 to Q.C.Z.), the Chinese Ministry of Science and Technology (Grant No. 2018YFA0107603 and 2019YFA0110002 to Q.C.Z.), and NIH NIAID grants (AI071068 and DK114483 to M.A.K.). Q.C.Z. thanks for support from the Beijing Advanced Innovation Center for Structural Biology and the Tsinghua-Peking Joint Center for Life Sciences.

## Author contributions

Q.J.L., M.A.K., and Q.C.Z. conceived of the project. J.Z. designed and performed sRNA icSHAPE-MaP experiments in 293T cells under the supervision of Q.C.Z. Q.J.L. and J.Z. designed and performed Dicer RIP-icSHAPE-MaP experiments under the supervision of M.A.K. and Q.C.Z. P.L., J.Z., Y.Z., and J.X. performed all bioinformatics analyses, supervised by Q.J.L. and Q.C.Z. Q.J.L. and Q.W. performed the experiments on sRNA-Seq for ~20 nt long RNAs, in vitro Dicer processing, and sRNA northern blots. B.R.C. performed the experiment of in vitro SHAPE analysis. Q.J.L., J.Z., P.L., M.A.K., and Q.C.Z. wrote the paper with input from all authors.

## Competing interests

The authors declare no competing interests.
