## [Peer Review File · Nature Communications]

Reviewers' comments:

Reviewer #1 (Remarks to the Author):

Luo et al. report a new approach to read out icSHAPE modifications, which are chemical probing marks of open and accessible bases on the RNA. The approach uses conditions for reverse transcription (addition of manganese) that allows the Superscript II enzyme to read through and introduce a mutation at the icSHAPE modified nucleotides in the RNA. The advantage of Luo et.al approach is coverage over the last 30nt at the 3'end of an RNA (the structure information on these 3'ends were lost in the original icSHAPE RT-stops based approach developed by Spitale et.al).

The authors then use their approach to capture the structures of small RNAs, and specifically RNAs that are substrates of Dicer. The major claim of the paper is that "Transcriptome-wide RNA structure probing reveals the structural basis of Dicer binding and cleavage." However, this claim has very little support and the data is not convincing. Here are my major concerns below:

- 1) The authors never show transcriptome-wide data. Indeed , they say in the methods they specifically isolate small RNAs , and thus provide chemical probing information only on RNAs that are less than 200nt- about 200 transcripts.
- 2) There is not enough data to confirm that this approach is accurate enough to generate 3-D models that lead to the major claim for the rule of physical distance (see also point 5) . The authors show that the tRNAs have AUC (area under curve) $> 0.5 < 0.8$, where 0.5 represents random data. In other words their data is only slightly better than random, and it is solely assessed on tRNAs (not as a general transcriptome-wide approach).
- 3) The authors claim their approach is "in vivo" (Fig.1A) but there is no description anywhere in the methods for adding icSHAPE directly to cells. In fact, the first time icSHAPE is mentioned is after the RIP protocol.
- 4) There is absolutely no evidence that RNA structures will remain the same through the entire pull down of Dicer, which involved cell lysis, harsh buffers and washing conditions. Indeed, I expect most of the RNA structures will be greatly affected by the RIP process. A far better experiment would be to do icSHAPE first, before the RIP.
- 5) The authors conclude that the distance of 60A is the ideal distance and explains why some miRNAs follow the loop counting rule vs the 5/3'counting (Fig.4, Supplementary Fig 4). I have two major problems with this claim:
 - a. If the authors didn't use the icSHAPE constrains, would they obtain very similar results for the 3D RNA structure predictions and distance measurements? In other words, are the icSHAPE constrains necessary? Typically, FARFAR/Rosetta approach perform extremely well on short RNAs and given all the issues with the icSHAPE data, it is highly unlikely these constrains give meaningful information to the 3D models.
 - b. Fig4.F is not compelling to draw a conclusion that 60A is the optimal distance with highest expression. There are a few mutants (e.g. U20A,U20C,WT) whose expression is clearly highest between 58-59.5A and drops a lot at 60.5A. The DM mutants have two fold difference in expression despite both having exactly 60A distance. The highlighted grey box is misleading.
- 6) Fig. 3 - We already know these three different classes of RNA molecules (snoRNA, tRNA, mRNA) have an intrinsic structure signature that is very obvious even if they weren't categorized as Dicer substrates. It is not surprising at all that they cluster into distinct groups. Fig.3 is currently not adding new information to the field. A far more useful conclusion would be picking up something unexpected or explaining the outliers.

Minor comments:

1. Dicer/RISC/Ago are not well introduced and their function is not sufficiently discussed.
2. Supplementary Fig.1b – authors should show Pearson correlation per transcript not just one number for all the transcripts.
3. Supplementary Fig.1h – it is irrelevant to compare the coverage needed for small RNA sequencing

(discussed here) vs transcriptome wide coverage on all RNAs including mRNAs.

4. Fig1D. – There is a big difference in the median signal of C vs G bases and the two should be equivalent. Is this due to a bias in the modification of icSHAPE or the RT read through strategy? It would have been much better to include a denatured control where the RNA is unfolded and all four bases are expected to react and be detected equally well.

Reviewer #2 (Remarks to the Author):

In this manuscript, Luo et al., optimized a previously established icSHAPE techniques for probing RNA structures and applied it to Dicer bound small RNAs. The main technical improvement is to ligate 5' and 3' adaptors to NAI-N3 treated RNA fragments (icSHAPE) and use mis-incorporation events to identify NAI-N3 modified nucleotides (ssRNA). By doing so, this mitigated the 3' drop-off problem of icSHAPE, which is particularly severe for small RNAs because of their short length.

Although this is a useful combination of the existing icSHAPE and MaP technique to probe the secondary structures of small RNAs (esp. miRNA), I have several concerns about the technical robustness of some of the results and to which extent this technique can significantly advance functional studies of miRNAs.

Specific comments:

1. The correlation between the icSHAPE-MaP score and base-paired nucleotides should be visualized and exceptions such as highly modified nucleotide within a stable double-stranded RNA region should be explored.

2. In vivo and in vitro icSHAPE-MaP results should be directly compared. Discrepancy should be analyzed if they are affected by RNA binding proteins or other in vivo contributors for such differences.

3. It is impressive that they can identify endogenously modified nucleotides in DMSO treated samples. However, in addition to Fig. S1F that show NAI-N3 treatment increases the percentage of mutated reads, they should quantify the overlap between mutated vs non-mutated nucleotides in DMSO vs NAI-N3 treated samples. In addition, Zubradt et al., has shown SSII is inferior to TGIRT by producing much more indel events which reduce the resolution of RNA structure determination. Do they have access to TGIRT to determine whether it can further improve their method?

4. Although Fig. S2D showed the boundary of two selected RNA fragments, their secondary structure determined by icSHAPE-MaP should be shown to validate if they generally form hairpin like structures. Furthermore, the size distribution of each small RNA classes bound by Dicer such as pre-miRNA, snoRNA, tRNA, mRNA fragments (Fig. S2F) should be shown separately. Typically, Dicer has a footprint of 60~70nt, the size of canonical pre-miRNAs. If much longer RNAs e.g. more than 120nt are recovered, how do they confirm they're bona fide Dicer bound RNA?

5. In Rybak-Wolf et al.'s paper, they used PAR-CLIP and a WT Dicer construct to identify Dicer bound RNAs. Based on the distribution of reads, Rybak-Wolf et al., found a large number of tRNA, vtRNA and miRNAs are bound by Dicer. In this study, they used a catalytic-dead Dicer construct in a Dicer null 293T cell line to identify Dicer bound RNAs. However, the reads mapped to miRNAs and vtRNA seem very low (~0.5% for miRNA, ~0.025% for vtRNA in Table S2) and rRNA counts for ~50% of the total reads. Even for pre-miRNAs, which should be the canonical substrates, the commonly detected RNAs between these two studies are only ~50%. These observations raised issues on whether the catalytic-dead Dicer mutant or their RIP technique can effectively recover bona fide Dicer substrates, and more importantly whether the detected pre-miRNAs are representative. At least, they should show the reads number mapped to each pre-miRNA, and check with existing miRNA profiles for 293T cells for whether their RIP-seq identify similar miRNAs proportionally to their expression levels. Discrepancies, if exist, should be carefully examined and explained.

6. The secondary structure of small RNAs is generally well defined because of relative short sequences. In general, bioinformatic prediction based on folding energy is reasonably accurate, when compared to experimentally defined structures. So it's not surprising icSHAPE-MaP determined structure generally agrees with prediction (Fig. 2D and 2E). It will be more interesting to identify RNAs with very (relatively) different icSHAPE-MaP determined structures vs bioinformatic prediction, and explore the underlying reason for the differences. If icSHAPE-MaP determined structures can be validated over predicted structures, it will serve as a good example for using icSHAPE-MaP, which is quite complicated and costly at this point.

7. The use of icSHAPE-MaP score for K-means clustering and PCA is quite interesting. Can they provide information which parameters are enriched in PC1 and PC2 and seemingly segregating pre-miRNA vs snoRNA vs tRNA (Fig. 3A-C)? What's the rationale to use 30nt on each side of the central loop position rather than using 40- or 50nt?

8. The cleavage score is entirely based on relative sequencing frequency of a given small RNA recovered from the catalytic-dead Dicer mutant vs WT Dicer. Independent validation of cleavage activity such as Dicing assays in Fig. 4 should be provided to support the claim "a direct correlation between Dicer binding affinity and cleavage activity". In addition, my concerns over whether the mutant Dicer and the RIP method can capture representative pre-miRNA profiles should be addressed as outlined above (#5).

9. Although the mutational analysis of mir-217 cleavage by Dicer is interesting, I am not sure about the connection to icSHAPE-MaP determined secondary structures and its functional significance.

Minor points:

Pg. 32, line 22, "The 9ul mutation buffer" should be "The 9ul RT buffer".

Reviewer #3 (Remarks to the Author):

In this manuscript, Luo, Zhang, Li et al., developed an approach called icSHAPE-MaP, which combines 2'-hydroxyl acylation and mutational profiling to probe intact RNA structures in vivo and in vitro. This technique solves the lack of resolution at the 3'-end of previous techniques like DMS-seq and icSHAPE, because it is based in measuring the mutational rate during reverse transcription at the site of chemically modified nucleotides. This feature allows the authors to infer the structure of even small RNA molecules. To capitalize in this technical advantage, the authors used this technique to determine the structure of the pre-miRNAs and other Dicer substrates expressed in HEK293T. Based on the structural results, the authors can group Dicer substrates in three different classes (pre-miRNAs, snoRNAs and tRNAs), where only the pri-miRNAs are actual efficient substrates of Dicer. Detailed analysis of the structure leads the authors to confirm that Dicer preferentially cuts at ~60 angstroms of the free end, regardless of the number of nucleotides and experimentally validated this conclusion using wild-type miR-217 and mutant variants.

Overall, this manuscript fills the gap from previous RNA structure techniques as it presents an approach that can determine the structure of small RNA fragments. Their solid analysis and validation of Dicer substrates further strengthen the existing rules of Dicer cleavage. Therefore, this paper is of broad interest and suitable publication in Nature Communications if the authors address the following comments:

Major concerns:

1.- icSHAPE-MaP on Dicer substrates is performed after Dicer pull-down, and hence Dicer is still bound (and protecting) the small RNAs. My concern is that Dicer-binding may modification hinder the accessibility of NAI-N3 to RNA and thus cause false-negative results. I encourage the authors to clarify this point or to demonstrate that NAI-N3-mediated modification is not affected by the presence of bound proteins.

2.- In figure 2E, the authors compare the experimentally inferred structure of miR-125 from the structure in miRbase. In my opinion, this analysis should be expanded to all the pre-miRNA analyzed in figure 3B. The authors could present this data as a heat map like figure 3B, where each line represents a miRNA and the color of each position represents the degree of departure from the theoretical structure. These results would highlight the importance of experimental determination of structure and pinpoint which type of structures are more difficult to predict.

Minor concerns:

3.- The authors observed a variety of substrates bound to Dicer, notably snoRNA, tRNA and pre-miRNA, as well as regions of some mRNAs. They claimed that "these findings suggest additional Dicer functions beyond miRNA biogenesis". However, the binding of other substrates beyond pre-miRNA could be an artifact resulting from Dicer over-expression. The authors should estimate the level of Dicer over-expression in HEK Dicer KO cells and compared it to the expression of Dicer in wild-type HEK293 cells.

Reviewer #1 (Remarks to the Author):

Luo et al. report a new approach to read out icSHAPE modifications, which are chemical probing marks of open and accessible bases on the RNA. The approach uses conditions for reverse transcription (addition of manganese) that allows the Superscript II enzyme to read through and introduce a mutation at the icSHAPE modified nucleotides in the RNA. The advantage of Luo et.al approach is coverage over the last 30nt at the 3'end of an RNA (the structure information on these 3'ends were lost in the original icSHAPE RT-stops based approach developed by Spitale et.al).

The authors then use their approach to capture the structures of small RNAs, and specifically RNAs that are substrates of Dicer. The major claim of the paper is that "Transcriptome-wide RNA structure probing reveals the structural basis of Dicer binding and cleavage." However, this claim has very little support and the data is not convincing. Here are my major concerns below:

1) The authors never show transcriptome-wide data. Indeed, they say in the methods they specifically isolate small RNAs, and thus provide chemical probing information only on RNAs that are less than 200nt- about 200 transcripts.

RESPONSE

We would first like to thank the reviewer for all the guidance about how to improve our paper. Regarding this comment specifically, we agree that although icSHAPE-MaP is a high-throughput method and can be applied to probe transcriptome-wide RNA structures; we did not provide transcriptome-wide data in our manuscript, so we have followed the suggestion and revised the title as "RNA structure probing reveals the structural basis of Dicer binding and cleavage". We would like however, to note that this technology is able to be applied to transcriptome-wide studies.

2) There is not enough data to confirm that this approach is accurate enough to generate 3-D models that lead to the major claim for the rule of physical distance (see also point 5). The authors show that the tRNAs have AUC (area under curve) $> 0.5 < 0.8$, where 0.5 represents random data. In other words, their data is only slightly better than random, and it is solely assessed on tRNAs (not as a general transcriptome-wide approach).

RESPONSE

We apologize for not explaining our results well, which may have caused misunderstanding regarding the accuracy of our approach. Our data showed a median value of AUC as 0.72 (**Figure 2D**), which is well above the level for random data (expected AUC = 0.5). This level of AUCs is considered as an indicator of good quality for RNA structure probing experiments by many other studies. For example, a previous study on Zika virus in *Cell Host & Microbe* (PMID: 30472207) considered AUC ≥ 0.83 as a very high agreement between experimental probing data and reference RNA structure models. In addition, another study which developed a RNA probing method called SHAPES in *RNA* (PMID: 25805860) considered AUC ≥ 0.80 as an indication of a high agreement, with AUC > 0.68 as a good agreement. And the recently developed Keth-seq, published in *Nat. Chem. Biol.* (PMID: 32015521), considered AUC ≥ 0.71 as a good agreement.

In addition to using tRNA secondary structures—for which we detected accurate matching with our icSHAPE-MaP data (AUC: 0.74 ~ 0.84) (**Response Figure 1A**)—we now followed the suggestion and have included the 5S rRNA secondary structure for comparison in the revised manuscript, and we found that our RNA structure probing data agrees with the model with an AUC = 0.825 (**Figure 1C** and **Response Figure 1B**). We would like to note that RNAs with reliable secondary structural models resolved from tertiary structures are limited and thus we cannot perform a transcriptome-wide assessment.

We trust the reviewer would agree that, together, these data collectively support the capacity of icSHAPE-MaP for accurately probing the secondary structures of RNAs.

Response Figure 1. icSHAPE-MaP accurately probes the secondary structures of RNAs. Structure models are resolved from the PDB database for the three select tRNAs (**A**) and the 5S rRNA (**B**, same as Figure 1C). The bases are colored by their icSHAPE-MaP scores. It is clear from the figure that the nucleotides with high scores are generally single-stranded, suggesting the high accuracy of icSHAPE-MaP.

3) The authors claim their approach is “in vivo” (Fig.1A) but there is no description anywhere in the methods for adding icSHAPE directly to cells. In fact, the first time icSHAPE is mentioned is after the RIP protocol.

RESPONSE

We apologize for the misunderstanding here, which may have been caused by a lack of clarity in presenting the method in the main text, as well as some missing information in the **METHODS**. We have now fixed this issue in the **RESULTS** (page 5 line 18): for *in vivo* probing of icSHAPE-MaP, NAI-N₃ was added directly to cells. We have also added a new section to the revised **METHODS** to further clarify this issue, which reads as follows:

*“HEK293T cells were treated with NAI-N₃ as previously described (Spitale et al. Nature 2015). Briefly, HEK293T cells were scrapped off culture dishes and washed with PBS. Then the cells were resuspended in 100 mM NAI-N₃ and incubated at 37°C on a Thermomixer at 1000 rpm for 5 min. The reaction was stopped by centrifugation at 2500g for 1 min at 4°C, and the supernatant was subsequently removed. The cells were resuspended in 250 µl PBS, and then 750 µl TRIzol LS Reagent was added for RNA extraction. The extracted RNA was size-selected on a 6% Urea-PAGE gel for 25~200 nt. The details of library construction are described in the following section **Construction of icSHAPE-MaP libraries**”.*

4) There is absolutely no evidence that RNA structures will remain the same through the entire pull down of Dicer, which involved cell lysis, harsh buffers and washing conditions. Indeed, I expect most of the RNA structures will be greatly affected by the RIP process. A far better experiment would be to do icSHAPE first, before the RIP.

RESPONSE

We thank the reviewer for giving us the opportunity to clarify some important points, and also for providing us the idea about doing icSHAPE first followed by RIP. However, based on the following points, we believe that it is advantageous to do RIP first and icSHAPE second, and not *vice versa*.

(1) If icSHAPE goes first, the probing reagent NAI-N₃ would modify proteins, thereby disrupting binding and subsequent pull-down by their antibodies. The reason behind this is that NAI-N₃ specifically reacts with the amine group of lysine and the hydroxyl group of serine, threonine, arginine and tyrosine (PMID: 30132655).

(2) RIP pulldown largely maintains RNA structures in a physiologically relevant setting. Supportive evidence for this comes from a study about a non-coding RNA 7SK by RIP discovered its interaction with the BAF chromatin remodeling complex to control gene transcription, published in *NSMB* (PMID 26878240).

5) The authors conclude that the distance of 60A is the ideal distance and explains why some miRNAs follow the loop counting rule vs the 5/3'counting (Fig.4, Supplementary Fig 4). I have two major problems with this claim:

a. If the authors didn't use the icSHAPE constrains, would they obtain very similar results for the 3D RNA structure predictions and distance measurements? In other words, are the icSHAPE constrains necessary? Typically, FARFAR/Rosetta approach perform extremely well on short RNAs and given all the issues with the icSHAPE data, it is highly unlikely these constrains give meaningful information to the 3D models.

RESPONSE

We followed the suggestion and have now conducted an analysis to compare pre-miRNA structures between the conditions with or without the icSHAPE-MaP constraints (see below for Response Figure 2, also included in the revised manuscript as **Figure 2F**). Briefly, for each miRNA, we predicted the structures of these pre-miRNAs with their icSHAPE-MaP scores as constraints, deemed "experimentally inferred structures". We then compared this experimentally inferred structure with the theoretical model from the miRBase database. In the result Response Figure 2, each row represents a pre-miRNA and each column is a nucleotide position; the grey boxes represent nucleotides with the same structures in both of their experimentally inferred and theoretical models, while boxes in other colors suggest different structures between two types of models.

As shown in the figure, most (78/100) pre-miRNA structures have at least one structurally different position between the experimentally inferred and theoretical models. Furthermore, 47 of all pre-miRNAs (n=100) have at least 5 structurally different positions. Interestingly, the structurally different positions were often found around the terminal loop region (~±5nt around the coordinate "0"). Overall, our experimentally inferred structures modeled a larger terminal loop (median size=9nt, Response Figure 3, also included in the revised manuscript as **Figure 2G**), compared to those from the theoretical model of miRBase (median size=6nt). Select examples were shown in Response Figure 4 (also included as **Figures 2E** and **S3H** in the

revised manuscript). The systematic differences arise from the pairing for some bases in the terminal loop in miRBase structural modeling, which was not supported by experimental probing in our work. We obtained the 3D structure of pre-miR-21 in the PDB database and found that this structural model shows a much larger terminal loop than the corresponding theoretical model in miRNA. Remarkably, this large terminal loop agrees with our experimentally inferred structure of pre-miR-21 (Response Figure 5). We recall that, RNA structures are generally more extended in cells than in solutions or by *in silico* predictions. This could be explained that computational predictions have a tendency to maximize the number of base-pairings to minimize the free energy, which may not necessarily be true because of cellular constraints. These comparisons clearly underscore the utility of experimentally determined information to help resolve the *in vivo* structure.

For FARFAR/Rosetta, we agree that they are very powerful for building atomic models for short motifs of less than 20 nucleotides in length. However, importantly, we would like to note that, these tools depend on secondary structural information for tertiary structure predictions (PMID: 20190761). Several studies have already demonstrated that experimental secondary structure probing data significantly improved the overall accuracy of secondary structure modeling/prediction, especially for longer RNAs (PMID: 15123812, 19109441). For example, Mathews et al (PMID: 15123812) incorporated chemical modification constraints into the prediction algorithm of RNA secondary structure, and improved the prediction accuracy for 16 sequences with known secondary structures, such as 5S rRNA in *E. coli* and *C. albicans*, and RNase P in *S. cerevisiae*, from 67% to 76% on average. In particular, for those RNA structures predicted with an accuracy of <40%, on the basis of minimal free energy alone, their accuracy was improved from 28% to 78% on average with the incorporation of structural constraints. Another study (PMID: 19109441) took advantage of the SHAPE information and showed that it improved the prediction sensitivity from 42.9% to 96.4% for the P546 domain of b13 group I intron, with an increase from 56.5% to 95.7% for HCV IRES and from 49.7% to 97.2% for 16S rRNA in *E. coli*.

For pre-miRNAs, we also observed that using different RNA secondary structural models from sequence-only predictions or from experimentally constrained modeling often results in different tertiary models of pre-miRNAs, and in turn leads to different results in Dicer cleavage distance calculation. For example, the cleavage distance for miR-27b-3p calculated from its 3D structures derived from the icSHAPE-MaP constrained secondary structures is about 60Å (Response Figure 6A), but the cleavage distance calculated from the 3D structures derived from the corresponding miRBase theoretical model is about 63.5 Å (Response Figure 6A). A similar scenario was observed for pre-miR-24-2, with its cleavage distance for its experimentally inferred structures is about 56Å (Response Figure 6B), whereas that from its theoretical miRBase model is 31Å (Response Figure 6B). Together, our present work shows that differences in secondary structure models affect the resultant 3D RNA structure models from FARFAR/Rosetta, and thus affect the measurement of Dicer cleavage distance.

Response Figure 2 (also included as Figure 2F in the revised manuscript). A heatmap highlighting differences between secondary structure models of pre-miRNAs with or without icSHAPE-MaP data as constraints. We used the terminal loop of the experimentally inferred structure as the coordinate reference. Each row represents a pre-miRNA; each column is a nucleotide, with the relative position of each nucleotide away from the center of its loop on the precursor. Note that 30-nt upstream and downstream the loop center are shown here, in which some nucleotides for a few pre-miRNAs may be missing, since they are not exactly 60-nt in length. And also note that the secondary structural model from miRBase is produced without experimental data as constraints. Purple boxes indicate that these nucleotides are single-stranded in the experimentally inferred structures but are base-paired in the structures from the miRBase, while yellow boxes indicate the opposite scenario. Green boxes indicate that these nucleotides are based-paired in both

structure models, but with different pairing counterparts. Grey boxes indicate those nucleotide positions with identical structures in both models, either single-stranded or base-paired with the same counterparts.

Response Figure 3 (also included as Figure 2G in the revised manuscript). A violin plot showing the length distribution of the terminal loop for pre-miRNAs based on the structural models from miRBase prediction or icSHAPE-MaP.

Response Figure 4 (Same as Figures 2E and S3H). The secondary structure models and icSHAPE-MaP scores of pre-miR-125a and pre-miR-19a. The structures on the top were modeled using the *RNAStructure* software with their icSHAPE-MaP scores as constraints. The structures on the bottom were obtained from the miRBase database.

Response Figure 5. The secondary structure models for pre-miR-21. The upper one was the theoretical structure model from the miRBase database; the middle one was modeled using the *RNAStructure* software

with their icSHAPE-MaP scores as constraints; the bottom was resolved from the PDB database. The highlight part indicates the one that overlaps with the structure model from the PDB database.

Response Figure 6. Comparison of the secondary structure models and their spatial 3' arm cleavage distance for select pre-miRNAs. (A) pre-mir-27b, (B) pre-mir-24-2. The secondary structures were modeled with icSHAPE-MaP constraints (left), or predicted by the miRBase database (right). The 3D structures were modeled by FARFAR/Rosetta, upon top 10 of which their spatial 3' arm cleavage distances were measured, with the median value presented.

b. Fig 4.F is not compelling to draw a conclusion that 60Å is the optimal distance with highest expression. There are a few mutants (e.g. U20A,U20C,WT) whose expression is clearly highest between 58-59.5Å and drops a lot at 60.5Å. The DM mutants have two fold difference in expression despite both having exactly 60Å distance. The highlighted grey box is misleading.

RESPONSE

We appreciate the reviewer's sharp eyes in pointing this out and help us on clarifying our findings. We revisited our data on dicing distance (D_{3p_miRNA}) and found that the D_{3p_miRNA} for most of pre-miRNAs that follow "3' counting rule" was around $59.2 \pm 1 \text{ \AA}$. Thus, strictly speaking, the preferred D_{3p_miRNA} we proposed is not exactly 60 \AA , but rather around 59.2 \AA with deviation of 1 \AA . And we have revised our manuscript to avoid potential confusion about this as follows.

"The dicing distance of the miRBase-annotated miRNAs were closer to $59.2 \pm 1 \text{ \AA}$ than the alternative +1 and -1 isoforms (Table S4), suggesting a distance of $\sim 59 \text{ \AA}$ as the physical basis for the Dicer cleavage site selection by the 5' and 3' counting rules."

6) Fig. 3 - We already know these three different classes of RNA molecules (snoRNA, tRNA, mRNA) have an intrinsic structure signature that is very obvious even if they weren't categorized as Dicer substrates. It is not surprising at all that they cluster into distinct groups. Fig.3 is currently not adding new information to the field. A far more useful conclusion would be picking up something unexpected or explaining the outliers.

RESPONSE

We thank the reviewer for this thoughtful suggestion to dig deeper and expand our findings. Regarding **Figure 3**, we would contend that it does add new information to the field: in addition to pre-miRNAs, we established that snoRNAs, tRNAs, and mRNAs are Dicer substrates, and demonstrated differential binding preferences and cleavage patterns using high-throughput sequencing. We have now validated selected snoRNAs as Dicer substrates with additional biochemical data as new panels for **Figures 3** and **S4** (also as **Response Figures 7, 8, and 9** below).

First, we show that Dicer is responsible for the stability of its bound snoRNAs found in our study. **Response Figure 7** shows the Dicer *in vitro* cleavage assay for six snoRNAs from **Figure 3C**. Interestingly, the snoRNA levels were greatly reduced in the presence of Dicer, similar to our findings with pre-miR-19a (the panel "Full length" in **Response Figure 7**). Some snoRNAs (e.g., snord5, snord37, and snord56) produced predominant 22-nt sRNAs, with additional minor RNA products of different sizes. To validate this result in cells, we did a qPCR experiment to quantify the expression level of five snoRNAs with Dicer complementation (see below for **Response Figure 8**). In general, their expression was significantly higher in NoDice cells, in which Dicer expression was deficient by the genome editing technology of TALEN nucleases (PMID: 24757167), compared to 293T cells. In the condition of WT Dicer complementation ("NoDice+WT Dicer"), snoRNA expression levels were greatly reduced, consistent with the data of the *in vitro* cleavage assay (**Response Figure 7**). Lastly, the expression levels for these snoRNAs were elevated in 293T cells in which endogenous Dicer expression was knocked-down with siRNAs (**Response Figure 9** below). Interestingly, we note that some of them also had reduced expression upon complementation with a mutant Dicer (**Response Figure 8**, "NoDice+Dead Dicer"), albeit to a lesser extent with WT Dicer, implying a regulatory role of Dicer in addition to its RNase activities.

Next, to find out whether the sRNA products by Dicer cleavage from these snoRNAs are routed to the miRNA pathway, we searched miRNA-Seq datasets from Bogerd *et. al.* (PMID: 24757167). We found that in sharp contrast to canonical miRNAs (e.g., miR-10a, miR-30c-2, and miR-92b), the sRNAs originating from select snoRNAs showed largely unaffected expression by Dicer deficiency, but dramatically reduced immunoprecipitation with AGO (**Response Figure 10**). To our understanding, the current general consensus is that RNAs must meet the following criteria to qualify as authentic miRNAs: (1) their sRNA production depends on Dicer; (2) their sRNAs load into Argonats (AGOs), the effectors of gene silencing. Therefore, the regulation of Dicer for these snoRNAs may not necessarily route them into the canonical miRNA pathway. Instead, Dicer may function as a gatekeeper in RNA metabolism for these and/or other bound substrates, a hypothetical function beyond the scope of canonical miRNA biogenesis.

We have now included the following discussions in the revised manuscript (from Page 12 Line 16) about Dicer substrate outliers and new discoveries from the structural clustering:

“We also detected Dicer substrates within mRNA introns and exons that display both high binding and cleavage activities (Table S3), including previously reported Dicer substrates such as the 5' UTR of glutamate-ammonia ligase (GLUL), exon 8 of aurora kinase B (AURKB), and H/ACA box 56 snoRNA (SNORA56) (Rybak-Wolf et al., 2014), validating our findings. Interestingly, some Dicer cleavage substrates share sequence homology to Alu elements, including intron 4 of the origin recognition complex subunit 5 (ORC5), an intron of the signal recognition particle 68 (SRP68), and an intron of ER lipid raft associated 1 (ERLIN1) (Table S3). This may extend the list of Alu units regulated by Dicer, which have been implicated in age-related macular degeneration and stem cell proliferation (Hu et al., 2012; Kaneko et al., 2011). In addition, we detected a Dicer substrate within an intergenic region (chr14:20630387-20630516) that belongs to a repeat family with sequence homology to pre-tRNA-Tyr. These findings suggest additional Dicer functions beyond miRNA biogenesis.”

And (from Page 13 Line 5)

“For further validation of these Dicer substrates and exploration of the regulatory role of Dicer, we performed *in vitro* Dicer cleavage assay for select snoRNAs (Figure 3E). Interestingly, these snoRNA levels were substantially reduced in the presence of Dicer, similar to our findings with pre-miR-19a (the panel “Full length” in Figure 3E). Some snoRNAs, e.g. snord5, snord37, and snord56, produced predominant 22-nt sRNAs, with additional minor RNA products in different sizes. To validate this result in cells, we did a qPCR experiment to quantify the expression level of five snoRNAs with Dicer complementation (Figure S4D). In general, their expression was higher in NoDice cells, in which Dicer expression was deficient, compared to 293T cells. In the condition of WT Dicer complementation (“NoDice+WT Dicer”), snoRNA expression levels were greatly reduced, consistent with the data in the *in vitro* cleavage assay (Figure 3E). Lastly, the expression levels for these snoRNAs were elevated in 293T cells in which endogenous Dicer expression was knocked-down with siRNAs (Figure S4E). Taken together, these lines of evidence suggest that Dicer is responsible for the stability of its bound snoRNAs found in our study.”

Response Figure 7 (also included as Figure 3F in the revised manuscript). *In vitro* dicing assay validates select snoRNAs as Dicer cleavage substrates.

Response Figure 8 (also included as Figure S4D in the revised manuscript). qPCR showing the expression level of select snoRNAs with Dicer complementation. “293T”: wild type HEK293T cell line; **“NoDice”:** Dicer deficient HEK293T cell line; **“NoDice+YFP”:** Dicer deficient HEK293T cell line overexpressing a plasmid of YFP using the same vector backbone as the groups **“NoDice+WT/Dead Dicer”;** **“NoDice+WT Dicer”:** Dicer deficient HEK293T cell line overexpressing a plasmid of wild type Dicer; **“NoDice+Dead Dicer”:** Dicer deficient HEK293T cell line overexpressing a plasmid of catalytic-dead Dicer. Data are mean \pm SD, n = 3 biological replicates.

Response Figure 9. (also included as Figure S4E in the revised manuscript) qPCR showing the expression level of select snoRNAs in 293T cells with Dicer knockdown by siRNAs. “Control”: HEK293T cells treated with Lipofectamine as the transfection reagent; **“siNT”:** scramble siRNAs. **“si-hDicer #1/#2/#3”:** three different siRNAs against human Dicer. Data are mean \pm SD, n = 3 technical replicates.

Response Figure 10. Expression of small RNAs associated with pre-miRNAs or snoRNAs. The expression level in the unit of RPM (read per million) was calculated with miRNA-Seq datasets from Bogerd et al. (PMID: 24757167). Fold change of expression of miRNAs and snoRNAs were presented between two different experiment settings. Left column: Dicer deficient cells (NoDice, PMID: 24757167) and 293T (HEK293T cells); right column: AGO-IP (Immunoprecipitation of RISC with an anti-Ago2 antibody) and 293T cells.

Minor comments:

1. Dicer/RISC/Ago are not well introduced and their function is not sufficiently discussed.

RESPONSE

We thank the reviewer for this suggestion on improving our manuscript. We have added an introduction for Dicer/RISC/Ago's functions, as follows (from Page 4 Line 13):

“Dicer belongs to the RNase III family and cleaves double-stranded RNAs (dsRNAs) and precursor microRNAs (pre-miRNAs) into mature small interfering RNAs (siRNAs) or microRNAs (miRNAs), respectively (Bernstein et al., 2001; Grishok et al., 2001; Hutvagner et al., 2001). The mature miRNAs/siRNAs load into Argonaute proteins to form a RISC complex, which represses target gene expression directed by Watson-Crick base-pairing between the guide strand and the target genes (Carthew and Sontheimer, 2009).”

2. Supplementary Fig.1b – authors should show Pearson correlation per transcript not just one number for all the transcripts.

RESPONSE

We thank the reviewer for this helpful suggestion on more properly presenting our result. We have revised **Figure S1B** to show Pearson correlation coefficient per transcript (see below for **Response Figure 11**). The violin plot shows that the correlation coefficient of mutation rate for most transcripts are higher than 0.95, showing the highly reproducibility of our experiments.

Response Figure 11 (also included as Figure S1B in the revised manuscript). Violin plot of Pearson correlation coefficient of mutation rates among replicates of *in vivo*, *in vitro* or DMSO groups.

3. Supplementary Fig.1h – it is irrelevant to compare the coverage needed for small RNA sequencing (discussed here) vs transcriptome wide coverage on all RNAs including mRNAs.

RESPONSE

We thank the reviewer for the question, but we suspect that this relates to a misunderstanding about **Figure S1I** (see below for **Response Figure 12**). The read coverage shown here belongs to the sliding windows (50 nt in length, 10 nt per step) on each of the targeted RNAs, rather than the overall coverage of the whole transcriptome. The sliding window strategy we used here set the length of regions of interest as a controlled variable, which is suitable for comparison of transcriptome or targeted RNAs with different lengths. In other words, the methodology shown here can be applied to the targeted RNA pool (either transcriptome or short length RNAs). Moreover, since DMS has its modification bias, we compared the mutation rates of A and C bases between these two methods, and our analysis clearly shows the advantages of icSHAPE-MaP over current technologies in terms of reproducibility and required sequencing cost.

Response Figure 12 (also included as Figure S1I in the revised manuscript). Accumulative distribution curve for Pearson correlation coefficients for mutation rates in DMS (left) or icSHAPE-MaP (right) samples for A and C bases for the average of two replicates. The six colored lines represent the cutoffs for different read coverage.

4. Fig1D. – There is a big difference in the median signal of C vs G bases and the two should be equivalent. Is this due to a bias in the modification of icSHAPE or the RT read through strategy? It would have been much better to include a denatured control where the RNA is unfolded and all four bases are expected to react and be detected equally well.

RESPONSE

We thank the reviewer for the comment and advice. Although C and G bases tend to form base pairing in RNA secondary structures, the mutation rates of these two bases are not necessarily equivalent (also for A and U), which are determined by their interference with the RT enzyme. We have followed the suggestion and compared our results with that from a denatured control (see below for Response Figure 13). We observed a similar trend of mutation rates for all four bases in both DMSO and the denatured group. Thus, we hypothesize that the RT read-through strategy may play a role here. Importantly, we did not find that the addition of the denatured group improved the accuracy of the SHAPE value calculation (Response Figure 14). Using 5S rRNA as an example, its AUC value was almost the same for secondary structure models, with or without the denatured condition. In conclusion, we posit that the DMSO group is sufficient to serve as a background control for the calculation of icSHAPE-MaP scores.

Response Figure 13. Box plot of mutation rates for all four RNA bases in different experiment groups.

Response Figure 14. icSHAPE-MaP scores with (A) or without (B) denatured conditions on the secondary structure models of 5S rRNA.

Let us take this opportunity to express our gratitude to the reviewer for the guidance about how to improve our study. Many thanks.

Reviewer #2 (Remarks to the Author):

In this manuscript, Luo et al., optimized a previously established icSHAPE techniques for probing RNA structures and applied it to Dicer bound small RNAs. The main technical improvement is to ligate 5' and 3' adaptors to NAI-N3 treated RNA fragments (icSHAPE) and use mis-incorporation events to identify NAI-N3 modified nucleotides (ssRNA). By doing so, this mitigated the 3' drop-off problem of icSHAPE, which is particularly severe for small RNAs because of their short length.

Although this is a useful combination of the existing icSHAPE and MaP technique to probe the secondary structures of small RNAs (esp. miRNA), I have several concerns about the technical robustness of some of the results and to which extent this technique can significantly advance functional studies of miRNAs.

Specific comments:

1. The correlation between the icSHAPE-MaP score and base-paired nucleotides should be visualized and exceptions such as highly modified nucleotide within a stable double-stranded RNA region should be explored.

RESPONSE

We thank the reviewer for acknowledging the utility of our study, for the helpful guidance generally and this suggestion specifically. We have conducted new analyses using tRNAs as examples to explore the correlation between icSHAPE-MaP scores and base-paired nucleotides (Response Figure 15). In general, we found that the icSHAPE-Map scores for double-stranded nucleotides (mean value = 0.430) were much smaller than those for single-stranded ones (mean value = 0.120) ($p < 10e-32$ with unpaired t-test). Very few exceptions existed for double-stranded nucleotides with high scores (>0.7), which were about 5% for all tRNAs.

We also explored highly modified nucleotide within stable double-stranded RNA regions and found that many of them may be related to endogenous modifications. For example, the highly modified 10G base of the tRNAs Asn-GTT-1-1 and Phe-GAA-1-1 are endogenously modified as N²-methylguanosine (m2G), of which a methyl group in the base-pairing interface may expose its 2' hydroxyl group and result in a higher reactivity to NAI-N3 (Response Figure 16, the left and middle panels). For the tRNA Ala-AGC-1-1, the highly modified 6U and 10G have also been suggested by a previous study to be endogenous modification sites, although the modification types are unclear (PMID: 28793268) (Response Figure 16, the right panel).

Response Figure 15 (also included as Figure S1E in the revised manuscript). icSHAPE-MaP score distribution of single-stranded (n=3,116) and double-stranded (n=4,226) nucleotides of tRNAs.

Response Figure 16. Correlation of the icSHAPE-MaP scores and endogenous modification sites (yellow or blue circles) in double-stranded regions of select tRNAs. The yellow colored nucleotides are m2G modification sites annotated in the MODOMICS database (PMID: 29106616) and the blue ones are those with uncertain modification types found by Gogakos et al. (PMID: 28793268).

2. *In vivo* and *in vitro* icSHAPE-MaP results should be directly compared. Discrepancy should be analyzed if they are affected by RNA binding proteins or other *in vivo* contributors for such differences.

RESPONSE

We thank the reviewer for this helpful suggestion about how to expand the scope of our study. In the following analysis, we directly compared the results between *in vivo* and *in vitro* icSHAPE-MaP. We used the absolute difference of icSHAPE-MaP scores of each transcript between *in vivo* and *in vitro* conditions for comparisons (see below for Response Figure 17A). A larger difference in icSHAPE-MaP scores may mean a bigger difference between their secondary structures. Based upon this assumption, overall snoRNAs show the biggest difference in icSHAPE-MaP scores between *in vivo* and *in vitro* conditions, a trend suggesting that their structures vary substantially between these two conditions. We note that this is also consistent with a previous finding showing that snoRNAs undergo the most dramatic change between *in vivo* and *in vitro* conditions among all observed non-coding RNAs (PMID: 25799993). tRNAs display higher consistency for their icSHAPE-MaP scores both *in vivo* and *in vitro* (Response Figure 17A). RNA binding proteins may contribute to the structural difference between *in vivo* and *in vitro* conditions (PMID: 30886404). For example, the box C/D snoRNAs form a specific kink-turn (K-turn) structure at C/D and C'/D' box regions, which are recognized by the K-turn binding protein, a protein required for assembly of snoRNPs (PMID: 17284456). The binding of K-turn binding protein is known to stabilize the K-turn structure during the initiation of snoRNP complex assembly *in vivo* (PMID: 17284456). Examples for the discrepancy between *in vivo* and *in vitro* groups are illustrated by the relatively higher icSHAPE-MaP scores in the C/D or C'/D' box region in the *in vitro* group compared to those in the *in vivo* group (see below for Response Figure 17B&C). This discrepancy between *in vivo* and *in vitro* groups may be due to the stabilization of k-turn structures by their binding proteins.

Response Figure 17. (A) (also included as **Figure S2** in the revised manuscript) Structure differences between *in vivo* and *in vitro* icSHAPE-MaP structure profiles defined as average absolute difference of icSHAPE-MaP scores of each nucleotide in one transcript. RNA are ranked by structure difference and colored by their type. (B and C) Examples of different icSHAPE-MaP scores of snoRNAs between the *in vivo* and *in vitro* groups. (B) SNORD95, (C) SNORD69. As a reference, the structure models shown here are from the Rfam database without icSHAPE-MaP constraints. C/D and C'/D' box regions are highlighted.

3. It is impressive that they can identify endogenously modified nucleotides in DMSO treated samples. However, in addition to Fig. S1F that show NAI-N3 treatment increases the percentage of mutated reads, they should quantify the overlap between mutated vs non-mutated nucleotides in DMSO vs NAI-N3 treated samples. In addition, Zubradt et al., has shown SSII is inferior to TGIRT by producing much more indel events which reduce the resolution of RNA structure determination. Do they have access to TGIRT to determine whether it can further improve their method?

RESPONSE

We thank the reviewer for the helpful suggestion. We have now quantified the overlap between mutated vs non-mutated nucleotides in DMSO vs NAI-N₃ treated samples, both *in vivo* and *in vitro* (**Response Figure 18A**). We choose a 0.002 as cutoff to exclude sequencing error in the definition of mutated nucleotides. At this cutoff, the nucleotides can be classified into four categories (“non-mutated”: those are not mutated (mutation rate < 0.002) in both the DMSO and NAIN3 groups; “both mutated”: those are mutated in both the DMSO and NAIN3 groups; those “only mutated in DMSO” group; and those “only mutated in NAIN3” group). Note that a substantial fraction (~32%, red region) of nucleotides are both mutated in DMSO and NAIN3 groups, while a large fraction (~36%, green region) of the nucleotides are mutated in the NAIN3 group

compared to a few (~2%, yellow region) were mutated only in the DMSO group. In other words, most of the nucleotides that are mutated in the DMSO group are also mutated in the NAIN3 group and the additional mutated nucleotides in the NAIN3 group are independent of DMSO.

For TGIRT, we did try the enzyme, but did not find its advantages over SSII in our experiments. Using the 5s rRNA for a direct comparison, the AUC of its structure determined by icSHAPE-MaP using SSII was much higher than that using TGIRT (0.825 Vs. 0.532. Response Figure 18B-C). Thus, we did not continue further with TGIRT.

Response Figure 18. (A) Stacked Barplots showing the overlap between mutated vs. non-mutated nucleotides. (B and C) icSHAPE-MaP scores of 5S rRNA using icSHAPE-MaP with SSII-Mn²⁺ (B) or TGIRT(C).

4. Although Fig. S2D showed the boundary of two selected RNA fragments, their secondary structure determined by icSHAPE-MaP should be shown to validate if they generally form hairpin like structures. Furthermore, the size distribution of each small RNA classes bound by Dicer such as pre-miRNA, snoRNA, tRNA, mRNA fragments (Fig. S2F) should be shown separately. Typically, Dicer has a footprint of 60~70nt, the size of canonical pre-miRNAs. If much longer RNAs e.g. more than 120nt are recovered, how do they confirm they're bona fide Dicer bound RNA?

RESPONSE

We thank the reviewer for this helpful comment and suggestion. We have now generated the secondary structural models by using icSHAPE-MaP data as constraints (Response Figure 19), which validate that they generally form hairpin like structures. Note that in addition to two selected RNA fragments in **Figure S2D** (**Figure S3D** in the revised manuscript), we now include two more examples of introns (from EML3 or CLK4, respectively).

In addition, **Figures S2E** and **S2F** (**Figures S3E** and **S3F** in the revised manuscript) have been revised to show the size distribution of each small RNA classes bound by Dicer, which shows that most substrate we found are indeed close to 60~70 nts (Response Figure 20). While we agree that length could be an evidence

to determine whether the RNA is a true substrate, we also would like to note that here we have enrichment scores of Dicer IP and a battery of other experiments, including *in vitro* Dicer cleavage assay, Dicer complementation, and siRNA perturbation of Dicer in cells, to verify whether select Dicer bound RNAs are its cleavage substrates (also please refer to our response to comment #8 below for all the details).

Response Figure 19 (also included as Figure S3D in the revised manuscript). Read coverage and secondary structure models determined by icSHAPE-MaP for select examples of intronic RNA fragments bound by Dicer.

Response Figure 20 (also included as Figure S3E in the revised manuscript). Size distribution for different types of Dicer-bound RNA fragments.

5. In Rybak-Wolf et al.'s paper, they used PAR-CLIP and a WT Dicer construct to identify Dicer bound RNAs. Based on the distribution of reads, Rybak-Wolf et al., found a large number of tRNA, vtRNA and miRNAs are bound by Dicer. In this study, they used a catalytic-dead Dicer construct in a Dicer null 293T cell line to identify Dicer bound RNAs. However, the reads mapped to miRNAs and vtRNA seem very low (~0.5% for

miRNA, ~0.025% for vtRNA in Table S2) and rRNA counts for ~50% of the total reads. Even for pre-miRNAs, which should be the canonical substrates, the commonly detected RNAs between these two studies are only ~50%. These observations raised issues on whether the catalytic-dead Dicer mutant or their RIP technique can effectively recover bona fide Dicer substrates, and more importantly whether the detected pre-miRNAs are representative. At least, they should show the reads number mapped to each pre-miRNA, and check with existing miRNA profiles for 293T cells for whether their RIP-seq identify similar miRNAs proportionally to their expression levels. Discrepancies, if exist, should be carefully examined and explained.

RESPONSE

We thank the reviewer for this thoughtful comment. The percentage of the shared pre-miRNAs identified in our RIP experiment and the PAR-CLIP experiment over the total number of our RIP result is 63.7%, well above 50% (**Supplemental Table 1** and **Response Table 1** below).

We have also added a new supplementary table to show the read number mapped to each pre-miRNA (**Supplemental Table 7**). In addition, we have checked our own miRNA-Seq data for 293T cells and have found that our RIP-seq identified similar miRNAs proportionally to their expression levels (see below for **Response Figure 21**). The Pearson correlation coefficient was 0.69 between these two datasets. This suggests that the catalytic-dead Dicer mutant or our RIP experiment effectively recovered *bona fide* Dicer substrates, supporting that the detected pre-miRNAs are generally representative.

Gene type	RIP	Overlap	Overlap Ratio	Cell 2014
pre-miRNA	273	174	63.74%	321
tRNA	151	83	54.97%	369
mRNAExon	204	91	44.61%	3808
lncRNAIntron	9	3	33.33%	85
miscRNA	65	18	27.69%	1140
lncRNAExon	13	3	23.08%	98
intergenic	126	24	19.05%	1238
snRNA	37	6	16.22%	30
mRNAIntron	481	67	13.93%	1237
snoRNA	216	14	6.48%	33

Response Table 1. Comparison of Dicer binding sites identified by RIP (this study) and PAR-CLIP (Rybak-Wolf et al, Cell, 2014).

Response Figure 21. Correlation of the expression levels of mature miRNA by miRNA-Seq in HEK293T cells and the abundance of precursor miRNAs by our RIP-seq.

6. The secondary structure of small RNAs is generally well defined because of relative short sequences. In general, bioinformatic prediction based on folding energy is reasonably accurate, when compared to experimentally defined structures. So it's not surprising icSHAPE-MaP determined structure generally agrees

with prediction (Fig. 2D and 2E). It will be more interesting to identify RNAs with very (relatively) different icSHAPE-MaP determined structures vs bioinformatic prediction, and explore the underlying reason for the differences. If icSHAPE-MaP determined structures can be validated over predicted structures, it will serve as a good example for using icSHAPE-MaP, which is quite complicated and costly at this point.

RESPONSE

We are sorry that we should have emphasized the differences between icSHAPE-MaP determined structures vs. bioinformatic predictions—initially we only included a few examples (e.g., **Figures 2E, S2H**). Following the guidance of this reviewer, we have now conducted an analysis to compare pre-miRNA structures between the conditions with or without the icSHAPE-MaP constraints (**Response Figure 22** below, also included in the revised manuscript as **Figure 2F**). Briefly, for each miRNA, we predicted its secondary structure with icSHAPE-MaP scores as constraints and named this structure as the “experimentally inferred structure”. We then compared this experimentally inferred structure with the theoretical model from the miRBase database. In the result **Response Figure 22**, each row represents a pre-miRNA and each column is a nucleotide position; the grey boxes represent nucleotides with structures the same in their experimentally inferred and theoretical models, while boxes in other colors suggest different structures between two types of models.

As shown in the figure, most (78/100) pre-miRNA structures have at least one structurally different position between the experimentally inferred and theoretical models. Furthermore, 47 of all pre-miRNAs (n=100) have at least 5 structurally different positions. Interestingly, the structurally different positions were often found around the terminal loop region ($\sim\pm 5$ nt around the coordinate “0”). Overall, our experimentally inferred structures modeled a larger terminal loop (median size = 9nt, **Response Figure 23**, also included in the revised manuscript as **Figure 2G**), compared to those from the theoretical model of miRBase (median size = 6nt). Select examples were shown in **Response Figure 24** (also included as **Figures 2E and S3H** in the revised manuscript). The systematic differences arise from the pairing for some bases in the terminal loop in miRBase structural modeling, which was not supported by experimental probing in our work. We obtained the 3D structure of pre-miR-21 in the PDB database and found that this structural model shows a much larger terminal loop than the corresponding theoretical model in miRBase. Remarkably, this large terminal loop agrees with our experimentally inferred structure of pre-miR-21 (**Response Figure 25**). We recall that, RNA structures are generally more extended in cells than in solutions or by *in silico* predictions. This could be explained that computational predictions have a tendency to maximize the number of base-pairings to minimize free energy, which may not necessarily be true because of cellular constraints. These comparisons clearly underscore the utility of experimentally determined information to help resolve the *in vivo* structure.

Response Figure 22 (also Figure 2F in the revised manuscript). A heatmap showing the difference between secondary structure models of pre-miRNAs with or without icSHAPE-MaP data as constraints. We used the terminal loop of the experimentally inferred structure as the coordinate reference. Each row represents a pre-miRNA; each column is a nucleotide, with the relative position of each nucleotide away from the center of its loop on the precursor. Note that 30-nt upstream and downstream the loop center is shown here, in which some nucleotides for a few pre-miRNAs may be missing, since they are not exactly 60-nt in length. In addition, the secondary structural model from miRBase is produced without experimental data as a constraint. Purple boxes indicate that these nucleotides are single-stranded in the experimentally inferred structures but are base-paired in the structures from the miRBase, while yellow boxes indicate the opposite scenario. Green boxes indicate that these nucleotides are base-paired in both structural models, but with different pairing counterparts. Grey boxes indicate those nucleotide positions with identical structures in both models, either single-stranded or base-paired with the same counterparts.

Response Figure 23 (also included as Figure 2G in the revised manuscript). A violin plot showing the length distribution of the terminal loop for pre-miRNAs based on the structural models from miRBase prediction or icSHAPE-MaP.

Response Figure 24 (also included as Figures 2E and S3H in the revised manuscript). The secondary structure models and icSHAPE-MaP scores of pre-miR-125a and pre-miR-19a. The structures on the top were modeled using the *RNAstructure* software with their icSHAPE-MaP scores as constraints. The structures on the bottom were obtained from the miRBase database.

Response Figure 25. The secondary structure models for pre-miR-21. The upper model was a theoretical structure model from the miRbase database; the middle model was generated using the *RNAstructure* software with their icSHAPE-MaP scores as constraints. The bottom model was resolved from

the PDB database. Green color highlights the agreement between experimentally inferred model and the structure model from the PDB database.

7. The use of icSHAPE-MaP score for K-means clustering and PCA is quite interesting. Can they provide information which parameters are enriched in PC1 and PC2 and seemingly segregating pre-miRNA vs snoRNA vs tRNA (Fig. 3A-C)? What's the rationale to use 30nt on each side of the central loop position rather than using 40- or 50nt?

RESPONSE

We thank for the reviewer for this insightful suggestion to dig into the clustering results. Here we show the heatmap of the weights of each position in PC1 and PC2 (see Figure 3A-B and Response Figure 26 below), from which PC1 can separate the pre-miRNAs vs snoRNAs vs tRNAs. Indeed, the weights of each position in PC1 clearly recapitulate the structural feature of pre-miRNAs with a nearly perfect base-paired stem and a relative large terminal loop (Response Figures 26B, median size = 9nt). Higher icSHAPE-MaP values in the central positions with lower ones in the flanking regions result in more negative PC1 values, which appears to be responsible for the separation of pre-miRNAs vs snoRNAs vs tRNAs (Response Figure 26A). For PC2, it shows a chessboard-like pattern with high and low icSHAPE-MaP values occurring in succession, characterizing the cloverleaf structure of tRNAs and separating them from snoRNAs (Response Figure 26B). The weights of PC2 on each position anti-correlate with the average icSHAPE-MaP scores of cluster III, in which the majority is tRNAs, so that many tRNAs are negative in the PC2 dimension.

We have added discussion on this into the revised manuscript as follows (from Page 11 Line 7):

“The substrate structures obtained from Dicer RIP experiments established three distinct groups by the K-means clustering, based on the top two principal components (Figure 3A). Interestingly, the weights of positions in Principal Component 1(PC1) of PCA recapitulate the structural feature of pre-miRNAs with a nearly perfect base-paired stem and a relatively large terminal loop (Figures 3B). Higher icSHAPE-MaP values in the central positions with lower ones in the flanking regions result in more negative PC1 values, which appears to be responsible for the separation of pre-miRNAs vs snoRNA vs tRNA(Figure 3A). While in PC2, it shows a chessboard-like pattern with high and low weights occurring in succession, characterizing the cloverleaf structure of tRNAs and separates them from snoRNAs. Accordingly, we found that each cluster was populated with one major type of RNA species: Cluster I was dominated by pre-miRNAs, with Cluster II by snoRNAs and Cluster III by tRNAs.”

The rationale to use 30nt on each side of the central loop position is because most pre-miRNAs are around 60-nt in length. Longer flanking regions will bring up the issue of missing values, which impairs the power of the PCA analysis.

Response Figure 26. (A) (also included as **Figure 3A** in the revised manuscript) Two-dimensional scatter plot of PC1 (10.2%) and PC2 (6.8%) in the PCA analysis using icSHAPE-MaP scores of 30-nt flanking each center loop. The dot color indicates its RNA category. Transparent dots suggest that they are not enriched by Dicer RIP. K-means clustering divides all RNAs into three clusters. (B) (also included as **Figure 3B** in the revised manuscript) Heatmap of the weights of each position for PC1 and PC2 in the PCA analysis.

8. The cleavage score is entirely based on relative sequencing frequency of a given small RNA recovered from the catalytic-dead Dicer mutant vs WT Dicer. Independent validation of cleavage activity such as Dicing assays in Fig. 4 should be provided to support the claim “a direct correlation between Dicer binding affinity and cleavage activity”. In addition, my concerns over whether the mutant Dicer and the RIP method can capture representative pre-miRNA profiles should be addressed as outlined above (#5).

RESPONSE

We thank for the reviewer for this helpful suggestion. As per this reviewer’s suggestion, we now add new biochemical data to validate the cleavage activity of Dicer is directly correlates with Dicer binding affinity (**Response Figures 27, 28, and 29**, also as new panels for **Figures 3F, S4D and S4E**).

Response Figure 27 shows the Dicer *in vitro* cleavage assay for six snoRNAs from **Figure 3C**. Interestingly, these snoRNA levels were greatly reduced in the presence of Dicer, similar to our findings with pre-miR-19a (the panel “Full length” in **Response Figure 27**). Some snoRNAs, e.g. snord5, snord37, and snord56, produced predominant 22-nt sRNAs, with additional minor RNA products in different sizes. In addition to this direct validation of the correlation of Dicer cleavage activity and Dicer binding, we further investigated the expression level of Dicer cleavage product is dependent on the binding by quantifying the expression level of five snoRNAs with Dicer complementation in cells using qPCR (**Response Figure 28**). In general, their expression was significantly higher in NoDice cells, in which Dicer expression was deficient by the genome editing technology of TALEN nucleases (PMID: 24757167), compared to 293T cells. In the condition of WT Dicer complementation (“NoDice+WT Dicer”), snoRNA expression levels were greatly reduced, consistent with the data in the *in vitro* cleavage assay (**Response Figure 27**). In addition, the

expression levels for these snoRNAs were elevated in 293T cells in which endogenous Dicer expression was knockeddown with siRNAs (**Response Figure 29**). Taken together, these lines of evidence suggest that Dicer binding is responsible for the stability of its substrate snoRNAs. Interestingly, some of them also had reduced expression upon complementation with a mutant Dicer (**Response Figure 28**, “NoDice+Dead Dicer”), albeit to a lesser extent with WT Dicer, implying a regulatory role of Dicer in addition to its RNase activities.

Finally, this reviewer’s last question on **whether the mutant Dicer and the RIP method can capture representative pre-miRNA profiles** has been addressed in our response to the comment #5 above.

Response Figure 27 (also included as Figure 3F in the revised manuscript). *In vitro* Dicing assay validates select snoRNAs as Dicer cleavage substrates.

Response Figure 28 (also included as Figure S4D in the revised manuscript). qPCR showing the expression level of select snoRNAs with Dicer complementation. “293T”: wild type HEK293T cell line; “NoDice”: Dicer deficient HEK293T cell line; “NoDice+YFP”: Dicer deficient HEK293T cell line overexpressing a plasmid of YFP using the same vector backbone as the groups “NoDice+WT/Dead Dicer”; “NoDice+WT Dicer”: Dicer deficient HEK293T cell line overexpressing a plasmid of wild type Dicer; “NoDice+Dead Dicer”: Dicer deficient HEK293T cell line overexpressing a plasmid of catalytic-dead Dicer. Data are means ± SD, n = 3 independent biological replicates.

Response Figure 29. (also included as Figure S4E in the revised manuscript) qPCR showing the expression level of select snoRNAs in 293T cells with Dicer knockdown by siRNAs. “Control”: HEK293T cells treated with Lipofectamine as the transfection reagent; “siNT”: scramble siRNAs. “si-hDicer #1/#2/#3”: three different siRNAs against human Dicer. Data are means \pm SD, n = 3 independent technical replicates.

9. Although the mutational analysis of mir-217 cleavage by Dicer is interesting, I am not sure about the connection to icSHAPE-MaP determined secondary structures and its functional significance.

RESPONSE

We apologize for not describing it clear for the connection between the icSHAPE-MaP experimentally determined secondary structures of miR-217 and its functional significance. We have shown that experimentally determined information can help resolve *in vivo* secondary structure in the above response to Comment #6. Specifically, we conducted icSHAPE-MaP analysis on mir-217 and its variants and used the structure information as constraints for secondary structure modeling (**Response Figure 30**). The secondary structure models were then used as input for 3D structure prediction using the FARFAR/Rosetta algorithm. We have showed in our analysis that, the 3D structural models of pre-miR-217 and their variants built from their corresponding icSHAPE-MaP determined secondary structures much more accurately predicted their dicing products than miRbase theoretical models, demonstrating the significance of using icSHAPE-MaP determined secondary structures for functional studies.

Response Figure 30. (also included as Figure 4D in the revised manuscript). sRNA-Seq for ~20-nt long RNAs indicates a consistency for Dicer cleavage-site selection on 3p arms of pre-miR-217

variants, based on their spatial distance. For each isoform (shown as a blunt-ended line), its relative expression level and cleavage distance were shown. The reactivities to NAI-N3, indicated by the *in vitro* SHAPE analysis, were shown in colored dots for select nucleotides. Red dots suggest a high reactivity to NAI-N3, while black dots suggest no reactivities, with purple dots suggesting median reactivities.

Minor points:

Pg. 32, line 22, "The 9ul mutation buffer" should be "The 9ul RT buffer".

RESPONSE

We apologize for the typo and thank the reviewer for pointing out it to improve our manuscript. This typo has been corrected. And we have checked out our newly submitted manuscript to avoid any other typos.

We would like to take this opportunity to express our thanks to the reviewer for the guidance about how to improve our study.

Reviewer #3 (Remarks to the Author):

In this manuscript, Luo, Zhang, Li et al., developed an approach called icSHAPE-MaP, which combines 2'-hydroxyl acylation and mutational profiling to probe intact RNA structures in vivo and in vitro. This technique solves the lack of resolution at the 3'-end of previous techniques like DMS-seq and icSHAPE, because it is based in measuring the mutational rate during reverse transcription at the site of chemically modified nucleotides. This feature allows the authors to infer the structure of even small RNA molecules. To capitalize in this technical advantage, the authors used this technique to determine the structure of the pre-miRNAs and other Dicer substrates expressed in HEK293T. Based on the structural results, the authors can group Dicer substrates in three different classes (pre-miRNAs, snoRNAs and tRNAs), where only the pri-miRNAs are actual efficient substrates of Dicer. Detailed analysis of the structure leads the authors to confirm that Dicer preferentially cuts at ~60 angstroms of the free end, regardless of the number of nucleotides and experimentally validated this conclusion using wild-type miR-217 and mutant variants.

Overall, this manuscript fills the gap from previous RNA structure techniques as it presents an approach that can determine the structure of small RNA fragments. Their solid analysis and validation of Dicer substrates further strengthen the existing rules of Dicer cleavage. Therefore, this paper is of broad interest and suitable publication in Nature Communications if the authors address the following comments:

Major concerns:

1.- icSHAPE-MaP on Dicer substrates is performed after Dicer pull-down, and hence Dicer is still bound (and protecting) the small RNAs. My concern is that Dicer-binding may modification hinder the accessibility of NAI-N3 to RNA and thus cause false-negative results. I encourage the authors to clarify this point or to demonstrate that NAI-N3-mediated modification is not affected by the presence of bound proteins.

RESPONSE

We thank the reviewer for the supportive assessment of our study and the helpful guidance. Regarding this comment specifically, in a previous study we have provided experimental evidence that NAI-N₃-mediated modification is largely unaffected by the presence of bound proteins (*Nature*; PMID: 25799993). Here, we performed the following analysis to show that the presence of bound proteins does not substantially impact NAI-N3 modifications.

1) Data analysis on transcriptome-wide structures shows that NAI-N3 modification has no significant difference on the known binding sites of RNA binding proteins (RBPs) from those randomly selected. To show this, we revisited our previous data on transcriptomic structures (Sun et al., 2019, PMID: 30886404). We then systematically analyzed the distribution of icSHAPE reactivity scores for a large dataset of RBP binding sites curated from the POSTAR2 database (<http://lulab.life.tsinghua.edu.cn/postar/>, Crosslinking and Immunoprecipitation (CLIP) studies of 67 RBPs in HEK293T cells, PMID: 30239819) and compared the result to that from randomly selected nucleotides on the same transcripts (**Response Figure 31**). We observed no significant difference for the icSHAPE scores between these two groups (mean value: 0.204 for RBP regions vs. 0.203 for random regions, p-value not significant by the unpaired t-test), suggesting that in general NAI-N3 modification is not affected by RBP binding.

2) Crystal and NMR structural studies demonstrated that the size of NAI-N₃ is small (~8Å), suggesting it has minimized spatial hindrance by RBPs and other molecules that interact with RNAs. This has been shown in two previous studies (Spitale et al., 2013, PMID 23178934; Spitale et al., 2015, PMID 25799993).

The first analysis used ribosomal RNAs within the ribosome complex (Spitale et al., 2015, PMID 25799993) and found that most structure-flexible bases were probed successfully by NAI-N₃ (see in the following Response Figure 32A, adapted from *Spitale et. al 2013*, PMID: 23178934); in other words, NAI-N₃ can modify all four RNA bases with free access, even though the ribosome structure is very compact and many nucleotides of rRNAs are buried inside.

The second analysis was done with the NMR structure of the Rbfox2 protein-RNA complex (see the following Response Figure 32B, adapted from *Spitale et. al 2015*, PMID 25799993). The upper panel shows the 3D structural model and the bottom panel indicates the differences between the *in vivo* and *in vitro* secondary structures. The Rbfox binding “flips out” the 2'-hydroxyl groups of the cognate RNA motif to the solvent-exposed surface of the complex (although the nucleotide C3 becomes buried upon such binding). The icSHAPE scores for the Rbfox binding motif were larger *in vivo* than those *in vitro* (positive values for the differential icSHAPE scores around the coordinate “0” on the x-axis), suggesting that Rbfox does not block the 2'-hydroxyl groups on its binding sites from NAI-N₃ access.

Response Figure 31. Box plot showing the lack of statistically significant differences between the icSHAPE reactivity scores at RBP binding sites and random positions on the same transcripts in humans (CLIP-seq data of 67 RBPs).

Adapted from (Spitale et al., 2012, *Nature Chemical Biology*)

Adapted from (Spitale et al., 2015, *Nature*)

Response Figure 32. (A) Gel shifts of NAI-N₃ probing and three-dimensional model with NAI-N₃ modifications for the *S. cerevisiae* 5S rRNA. (adapted from *Spitale et. al 2013, PMID: 23178934*). (B) A three-dimensional model of Rbfox binding to RNA, and differences between *in vivo* vs. *in vitro* icSHAPE reactivity scores at Rbfox binding sites. (adapted from *Spitale et. al 2015, PMID: 25799993*).

2.- In figure 2E, the authors compare the experimentally inferred structure of miR-125 from the structure in miRbase. In my opinion, this analysis should be expanded to all the pre-miRNA analyzed in figure 3B. The authors could present this data as a heat map like figure 3B, where each line represents a miRNA and the color of each position represents the degree of departure from the theoretical structure. These results would highlight the importance of experimental determination of structure and pinpoint which type of structures are more difficult to predict.

RESPONSE

We highly appreciate this very helpful suggestion. Following the guidance of this reviewer, we have now conducted an analysis to compare pre-miRNA structures between the conditions with or without the icSHAPE-MaP constraints (**Response Figure 33**, also included in the revised manuscript as **Figure 2F**). Briefly, for each miRNA, we predicted its secondary structures with icSHAPE-MaP scores as constraints and named this structure as the “experimentally inferred structure”. We then compared this experimentally inferred structure with the theoretical model from the miRBase database. In the result **Response Figure 33**, each row represents a pre-miRNA and each column is a nucleotide position; the grey boxes represent nucleotides with structures the same in their experimentally inferred and theoretical models, while boxes in other colors suggest different structures between two types of models.

As shown in the figure, most (78/100) pre-miRNA structures have at least one structurally different position between the experimentally inferred and theoretical models. Furthermore, 47 of all pre-miRNAs (n=100) have at least 5 structurally different positions. Interestingly, the structurally different positions were often found around the terminal loop region (~±5nt around the coordinate “0”). Overall, our experimentally inferred structures modeled a larger terminal loop (median size = 9nt, **Response Figure 34**, also included in the revised manuscript as **Figure 2G**), compared to those from the theoretical model of miRBase (median size = 6nt). Select examples were shown in **Response Figure 35** (also included as **Figures 2E** and **S3H** in the revised manuscript). The systematic differences arise from the pairing for some bases in the terminal loop in miRBase structural modeling, which was not supported by experimental probing in our work. We obtained the 3D structure of pre-miR-21 in the PDB database and found that this structural model shows a much larger terminal loop than the corresponding theoretical model in miRBase. Remarkably, this large terminal loop agrees with our experimentally inferred structure of pre-miR-21 (**Response Figure 36**). We recall that, RNA structures are generally more extended in cells than in solutions or by *in silico* predictions. This could be explained that computational predictions have a tendency to maximize the number of base-pairings to minimize free energy, which may not necessarily be true because of cellular constraints. These comparisons clearly underscore the utility of experimentally determined information to help resolve the *in vivo* structure.

Response Figure 33 (also included as Figure 2F in the revised manuscript). A heatmap showing the difference between secondary structure models of pre-miRNAs with or without icSHAPE-MaP scores as constraints. We used the terminal loop of the experimentally inferred structure as the coordinate reference. Each row represents a pre-miRNA; each column is a nucleotide, with the relative position of each nucleotide away from the center of its loop on the precursor. Note that 30-nt upstream and downstream the loop center is shown here, in which some nucleotides for a few pre-miRNAs may be missing, since they are not exactly 60-nt in length. And also note that the secondary structural model from miRBase is produced without experimental data as a constraint. Purple boxes indicate that these nucleotides are single-stranded in the experimentally inferred structures but are base-paired in the structures from the miRBase, while yellow boxes indicate the opposite scenario. Green boxes indicate that these nucleotides are base-paired in both structural models, but with different pairing counterparts. Grey boxes indicate those nucleotide positions with identical structures in both models, either single-stranded or base-paired with the same counterparts.

Response Figure 34 (also included as Figure 2G in the revised manuscript). A violin plot showing the length distribution of the terminal loop for pre-miRNAs based on the structural models from miRBase prediction or icSHAPE-Map.

Response Figure 35 (also included as Figures 2E and S3H in the revised manuscript). The secondary structure models and icSHAPE-Map scores of pre-miR-125a and pre-miR-19a. The structures on the top were modeled using the *RNAstructure* software with their icSHAPE-Map scores as constraints. The structures on the bottom were obtained from the miRBase database.

Response Figure 36. The secondary structure models for pre-miR-21. The upper one was theoretical structure model from the miRbase database. The middle one was modeled using the *RNAstructure* software with their icSHAPE-Map scores as constraints. The bottom was resolved from the PDB database. The highlight part indicates the one that overlaps with the structure model from the PDB database.

Minor concerns:

3.- The authors observed a variety of substrates bound to Dicer, notably snoRNA, tRNA and pre-miRNA, as well as regions of some mRNAs. They claimed that “these findings suggest additional Dicer functions beyond miRNA biogenesis”. However, the binding of other substrates beyond pre-miRNA could be an artifact resulting from Dicer over-expression. The authors should estimate the level of Dicer over-expression in HEK Dicer KO cells and compared it to the expression of Dicer in wild-type HEK293 cells.

RESPONSE

We thank the reviewer for the guidance. Following the reviewer’s suggestion, we have now estimated the level of Dicer over-expression in HEK Dicer deficient cells and compared it to the expression of Dicer in wild-type HEK293T cells (see below for Response Figure 37). The over-expression level of catalytic-dead Dicer is about 10 times that of Dicer in the wild-type HEK293T cells, quantified by western blot. This level of Dicer expression combined with the non-crosslink RIP method is a trade-off between sensitivity and specificity on the identification of Dicer substrates.

We understand the possible concern that the Dicer substrate we discover are artificial results of its overexpression. We have thus taken several experimental and computational means to balance the sensitivity and specificity on discovering Dicer substrates. We calculated Dicer enrichment scores as an indication of binding affinity with Dicer, by comparing the transcript level of Dicer IP to that of input with RNA-Seq (Response Figure 38A). As seen, the enrichment score of some substrates are comparable to those for pre-miRNAs (Figure 3C and Response Figure 38A). In addition, we have measured Dicer’s cleavage activities in the background of its deficiency using RNA-Seq. A similar trend was evident in the Dicer cleavage scores for the same transcripts (e.g. box C/D snoRNAs) (Response Figure 38B), indicating that they are *bona fide* Dicer substrates, instead of an artifact resulting from its over-expression.

In addition, we added new experimental data to validate select snoRNAs as Dicer substrates. 1) Dicer *in vitro* cleavage assay shows their level greatly reduced to the similar extent as pre-miR-19a does in the presence of Dicer, validating them as Dicer cleavage substrates (Response Figure 39). 2) qPCR experiment shows their expression was higher in Dicer deficient cells (“NoDice” group in Response Figure 40), reduced when complemented with WT Dicer (“NoDice+WT Dicer” group in Response Figure 40), elevated when endogenous Dicer was knockdowned(Response Figure 41), suggesting they are Dicer’s substrates and are regulated by Dicer.

Taken these together, we take the view that Dicer may bind to a variety of substrates identified in our study, including snoRNAs, tRNAs and pre-miRNAs, as well as regions of some mRNAs; this view is also supported by a previous report (PMID: 25416952) that Rybak-Wolf *et al.* also reported some snoRNAs, tRNAs, and mRNA fragments are bound by Dicer using PAR-CLIP. Notably, the Dicer substrates we identified in the present study share substantial overlap with theirs (see below for Response Table 2).

Response Figure 37. The expression of Dicer in our study. (A) Western blots showed Dicer expression of four biological replicates for each group, including wild-type HEK293T cells (293T), HEK Dicer deficient cells (NoDice), and Dicer deficient cells with Dicer over-expression (Dicer-OX). Note that the Dicer over-expression was achieved using the same conditions described in Figure S3A. (B) Quantification of western blot data in panel A. Data are mean \pm SD, n = 4 biological replicates.

Response Figure 38. Violin plots showing (A) the Dicer enrichment scores and (B) cleavage scores for different classes of RNA (pre-miRNA, snoRNA, tRNA, mRNA fragments, etc.) of Dicer substrates identified by RIP in panel A or RNA-Seq in panel B.

Response Figure 39(also as Figure 3F in the new main text). *In vitro* dicing assay validates select snoRNAs as Dicer cleavage substrates.

Response Figure 40 (also as Figure S4D in the revised manuscript). qPCR showing the expression level of select snoRNAs with Dicer complementation. “293T”: wild type HEK293T cell line; “NoDice”: Dicer deficient HEK293T cell line; “NoDice+YFP”: Dicer deficient HEK293T cell line overexpressing a plasmid of YFP using the same vector backbone as the groups “NoDice+WT/Dead Dicer”; “NoDice+WT Dicer”: Dicer deficient HEK293T cell line overexpressing a plasmid of wild type Dicer; “NoDice+Dead Dicer”: Dicer deficient HEK293T cell line overexpressing a plasmid of catalytic-dead Dicer. Data are mean \pm SD, n = 3 biological replicates.

Response Figure 41 (also as Figure S4E in the revised manuscript). qPCR showing the expression level of select snoRNAs in 293T cells with Dicer knockdown by siRNAs. “Control”: HEK293T cells treated with Lipofectamine as the transfection reagent; “siNT”: scramble siRNAs. “si-hDicer #1/#2/#3”: three different siRNAs against human Dicer. Data are mean \pm SD, n = 3 technical replicates.

Gene type	RIP	Overlap	Overlap Ratio	Cell 2014
pre-miRNA	273	174	63.74%	321
tRNA	151	83	54.97%	369
mRNAExon	204	91	44.61%	3808
lncRNAIntron	9	3	33.33%	85
miscRNA	65	18	27.69%	1140
lncRNAExon	13	3	23.08%	98
intergenic	126	24	19.05%	1238
snRNA	37	6	16.22%	30
mRNAIntron	481	67	13.93%	1237
snoRNA	216	14	6.48%	33

Response Table 2. Comparison of Dicer binding sites identified by RIP (this study) and PAR-CLIP (Rybak-Wolf et al, Cell, 2014).

Let us again take this opportunity to express our gratitude to the reviewer for the supportive comments and helpful guidance about how to improve our study. Many thanks.

REVIEWERS' COMMENTS

Reviewer #1 (Remarks to the Author):

I do appreciate the substantial revisions added by the authors that have significantly improved the manuscript and addressed my major concerns. In sight of these revisions, I would recommend the manuscript for publication in Nature Communications.

Reviewer #2 (Remarks to the Author):

In the revised manuscript, Luo et al. have substantially improved this study. Many new data, including analysis and experiments, have clarified many of my previous concerns – such as the new Dicing data (Fig. 3F) and Dicer rescue (Fig. S4D, E), and new analysis (Fig. 2F). I only have two remaining issues.

1. They calculated the icSHAPE-MaP reactivity score based on the mutation rate in NAI vs DMSO samples. Although this score is “negatively correlates with the likelihood of the nucleotide being paired”, it will be useful if they can define the specific score range for double-stranded vs single-stranded regions as well as the potential score range for ambiguous regions e.g. regions that could be single- or double-stranded.

2. In new Fig. S2, they demonstrated that in vivo and in vitro icSHAPE-MaP differ, sometimes substantially. However, in their Dicer-bound transcript test (pg. 8, line 17), they performed NAI-N3 modification after the RNA enrichment. Why not treat the cells with NAI-N3 before Dicer RIP?

Reviewer #3 (Remarks to the Author):

In this extensive revision of the manuscript, Luo, Zhang, Li et al., have addressed the concerns that I raised during the first revision of the manuscript. The authors have reanalyzed the data or performed additional experiments to ease these concerns.

In particular, now the authors:

A.- provide evidence that the RNA-protein interactions do not preclude the action of NAI-N3. They show that the reactivity of known RNA-binding protein sites does not differ from other random regions and refer to previous literature to show other examples with known 3D structures where the RBP does not prevent the chemical modification of the RNA.

B.- provide a new figure that clearly shows the magnitude of their findings. New Figure 2F now shows all the miRNA analyzed in their experiments, all in a single heat map that allows to very easily visualize the fact that the authors experimentally validated structure of pre-miRNAs differs from the in silico predictions. Up to 47 out of 100 miRNAs have at least 5 structural changes compared to the annotated structure in miRbase, and 78 out of 100 have at least one permutation. These results convey the necessity of this type of experiments to correct the publicly available databases.

C.- provide an experimental estimation of the amount of Dicer overexpression. While a 10-fold overexpression may still be grounds to be concerned about unspecific binding to RNAs, the authors show that some of the non-miRNA species that bind to Dicer can be processed by Dicer in an in vitro assay. This result, together with the benchmarking with other published Dicer pull-downs, increase the confidence on the specificity of the Dicer pull-down.

Overall, I consider that the new technique that the authors develop to determine the structure of small RNAs, together with their findings on miRNA structure and Dicer substrate preferences, merit

publication and are of interest for the broad audience of Nature Communications. The authors should address the following minor concerns to enhance the manuscript:

1) The colors of the new Figure 2F are clearer in the rebuttal letter than in the manuscript. The authors should use the color code of the figure in the rebuttal letter.

2) The authors show in Figure 4D the structure of pre-miR-217 with a 3 nucleotide 3'-overhang. This configuration is likely not compatible with a Drosha cleavage, the enzyme responsible for doing this end of the stem. MirGeneDB shows sequencing data providing evidence that in some instances, pre-miR-217 5'-arm starts a nucleotide earlier, that would then generate the canonical 2 nt 3'-overhang. The authors should comment on this issue.

REVIEWERS' COMMENTS

Reviewer #1 (Remarks to the Author):

I do appreciate the substantial revisions added by the authors that have significantly improved the manuscript and addressed my major concerns. In sight of these revisions, I would recommend the manuscript for publication in Nature Communications.

Response:

We thank this reviewer for the ongoing support of our manuscript.

Reviewer #2 (Remarks to the Author):

In the revised manuscript, Luo et al. have substantially improved this study. Many new data, including analysis and experiments, have clarified many of my previous concerns – such as the new Dicing data (Fig. 3F) and Dicer rescue (Fig. S4D, E), and new analysis (Fig. 2F). I only have two remaining issues.

1. They calculated the icSHAPE-MaP reactivity score based on the mutation rate in NAI vs DMSO samples. Although this score is “negatively correlates with the likelihood of the nucleotide being paired”, it will be useful if they can define the specific score range for double-stranded vs single-stranded regions as well as the potential score range for ambiguous regions e.g. regions that could be single- or double-stranded.

Response:

We thank the reviewer for the helpful suggestion. Guided by this, we have now analyzed the distribution of icSHAPE-MaP reactivity scores in known single-stranded and double-stranded regions using two classes of non-coding RNAs, 5S rRNA and tRNAs (**Revision Figure 1A-B**). As expected, in general, single-stranded regions have much higher icSHAPE-MaP reactivity scores than the double-stranded regions (median scores: 0.523 vs 0.09 for 5S rRNA, 0.306 vs 0.043 for tRNAs). We would like to note that it is often ambiguous when seeking to define a clear cutoff to distinguish single-stranded regions from double-stranded ones. There are two main reasons for this. First, as this reviewer indicated, many RNA regions are actually ambiguous in structure (*i.e.*, capable of switching between single- and double-stranded forms in cells). Second, all probing methods inevitably have some level of noise in their high-throughput data. Nevertheless, we have now estimated icSHAPE-MaP reactivity scores and calculated its tendencies as single-stranded or double-stranded regions (**Revision Figure 1C**). As you can see in **Revision Figure 1C**, we estimate that icSHAPE-MaP reactivity scores less than 0.1 indicate double-stranded regions,

with the ones larger than 0.7 suggesting single-stranded regions, and the ones between 0.1~0.7 more likely to be ambiguous regions.

For comparison, we performed the same analysis using reactivity scores from previous studies of icSHAPE (PMID: 27180905) and DMS-seq (PMID: 24336214). We found that icSHAPE-MaP performs similarly well or even better than current technologies in separating single- and double-stranded regions (**Revision Figure 2**). The median of reactivity scores by icSHAPE-MaP were 0.523 for single-stranded regions versus 0.090 and 0.043 for double-stranded regions for 5S rRNA, and 0.306 versus 0.043 for tRNAs (**Revision Figure 1A-B**), while it was 0.069 versus 0.019 by icSHAPE for human 18S rRNA (**Revision Figure 2A**) or 0.100 versus 0.053 by DMS-seq for yeast 16S rRNA (**Revision Figure 2B**).

Revision Figure 1. The distribution of icSHAPE-MaP reactivity scores in single- or double-stranded regions of 5S rRNA and tRNAs. (A, B) The distribution of icSHAPE-MaP reactivity scores in single- or double-stranded regions of 5S rRNA (A) and tRNAs (B, Supplementary Figure 1e in the main text), p values were determined with a two-sided two-sample Student's t-test. Violin plots show the median (white circle), 25/75 percentiles, and smallest/largest values within 1.5 times the interquartile range above/below quartiles. (C) The smoothed percentage of single-stranded regions across different icSHAPE-MaP score ranges. The empirical score range for single-stranded, double-stranded, or ambiguous regions are indicated by boxes in different colors.

Revision Figure 2. The distribution of structure scores in single- or double-stranded regions of ribosomal RNAs from previous studies. The scores of icSHAPE reactivity (A) and DMS-seq (B) were calculated according to the description of previous papers (icSHAPE [PMID: 27180905] and DMS-seq [PMID:

24336214]). The human 18S rRNA and the yeast 16S rRNA reference structure models were collected from the CRW database (PMID: 11869452). Note that DMS-seq can only detect A/C bases. p values were determined with a two-sided, two-sample Student's t-test. Violin plots show the median (white circle), 25/75 percentiles, and smallest/largest values within 1.5 times the interquartile range above/below the quartiles.

2. In new Fig. S2, they demonstrated that *in vivo* and *in vitro* icSHAPE-MaP differ, sometimes substantially. However, in their Dicer-bound transcript test (pg. 8, line 17), they performed NAI-N₃ modification after the RNA enrichment. Why not treat the cells with NAI-N₃ before Dicer RIP?

Response:

We thank the reviewer for giving us the opportunity to clarify some important points. In our experiments, the required order of experiments was RIP first and icSHAPE second; this is because, if using icSHAPE first, the probing reagent NAI-N₃ would have modified proteins, thereby disrupting binding and subsequent pull-down by their antibodies. The reason behind this is that NAI-N₃ specifically reacts with the amine group of lysine residues and the hydroxyl group of serine, threonine, arginine, and tyrosine residues (PMID: 30132655).

We understand the concern here, and we have performed the RIP pulldown using a mild procedure which has been assumed to maintain RNA structures in a physiologically relevant setting (PMID 26878240). In that study, Flynn *et al.* pulled-down a non-coding RNA 7SK using RIP followed by icSHAPE structure probing. In fact, in our analysis, the icSHAPE-MaP scores of 5S rRNA between *in vivo* and RIP pulldown sample indeed are very similar (**Revision Figure 3**).

Revision Figure 3. Comparison of icSHAPE-MaP scores between *in vivo* and RIP pulldown sample on 5S rRNA. Pearson Correlation of icSHAPE-MaP scores is 0.78.

We again thank the reviewer for the helpful guidance and the ongoing support of our manuscript.

Reviewer #3 (Remarks to the Author):

In this extensive revision of the manuscript, Luo, Zhang, Li et al., have addressed the concerns that I raised during the first revision of the manuscript. The authors have reanalyzed the data or performed additional experiments to ease these concerns.

In particular, now the authors:

A.- provide evidence that the RNA-protein interactions do not preclude the action of NAI-N3. They show that the reactivity of known RNA-binding protein sites does not differ from other random regions and refer to previous literature to show other examples with known 3D structures where the RBP does not prevent the chemical modification of the RNA.

B.- provide a new figure that clearly shows the magnitude of their findings. New Figure 2F now shows all the miRNA analyzed in their experiments, all in a single heat map that allows to very easily visualize the fact that the authors experimentally validated structure of pre-miRNAs differs from the in silico predictions. Up to 47 out of 100 miRNAs have at least 5 structural changes compared to the annotated structure in miRbase, and 78 out of 100 have at least one permutation. These results convey the necessity of this type of experiments to correct the publicly available databases.

C.- provide an experimental estimation of the amount of Dicer overexpression. While a 10-fold overexpression may still be grounds to be concerned about unspecific binding to RNAs, the authors show that some of the non-miRNA species that bind to Dicer can be processed by Dicer in an in vitro assay. This result, together with the benchmarking with other published Dicer pull-downs, increase the confidence on the specificity of the Dicer pull-down.

Overall, I consider that the new technique that the authors develop to determine the structure of small RNAs, together with their findings on miRNA structure and Dicer substrate preferences, merit publication and are of interest for the broad audience of Nature Communications. The authors should address the following minor concerns to enhance the manuscript:

1) The colors of the new Figure 2F are clearer in the rebuttal letter than in the manuscript. The authors should use the color code of the figure in the rebuttal letter.

Response:

We thank the reviewer for the excellent summary and the ongoing support of our manuscript. Regarding this comment specifically, we have now replaced Figure 2F with the updated color code (the one in the rebuttal letter).

2) The authors show in Figure 4D the structure of pre-miR-217 with a 3 nucleotide

3'-overhang. This configuration is likely not compatible with a Drosha cleavage, the enzyme responsible for doing this end of the stem. MirGeneDB shows sequencing data providing evidence that in some instances, pre-miR-217 5'-arm starts a nucleotide earlier, that would then generate the canonical 2 nt 3'-overhang. The authors should comment on this issue.

Response:

To our understanding, there are actually two potential pre-miR-217 annotations in the MirGeneDB database, with one (hsa-Mir-217-v1) the same as our models (Figure 4D and **Revision Figure 4**) and the other one (hsa-Mir-217-v2) starting a nucleotide earlier. We have now aligned together the major isomiR of 5p (colored in black) and 3p (colored in red), we found that the 3' end of pre-miR-217 presents the 2-nt 3' overhang, a classical feature with Drosha cleavage (**Revision Figure 4**). We would also like to note that our deep sequencing data for sRNAs of ~20nt long shows that isomiRs corresponding to the 5' end of pre-miR-217-V2 (green blunt-ended lines) generally have very low expression comparing to the isomiRs corresponding to the 5' end of pre-miR-217-V1 (Black blunt-ended lines in **Revision Figure 4**). Previously, we only showed the major isoforms of miRNAs (isomiRs) on 5p or 3p arms in Figure 4D; now, we have replaced these with the content presented in **Revision Figure 4** to minimize a potential confusion on this point.

In addition, we agree with the reviewer that (taking a canonical view) Drosha functions as an RNase III and generally produces 2-nt 3'-overhangs of pre-miRNAs. This function relies on intrinsic features (including sequence and structure) for the pre-miRNAs and for Drosha itself. We would like to note that emerging evidence is now starting to indicate that alternative Drosha cleavage sites may exist for pre-miRNAs, raising the possibility that the canonical view of Drosha function(s) may need to be expanded. For example, with pre-miR-16-1, Drosha has been shown to produce primarily 1-nt 3'-overhangs instead of 2-nt overhangs (PMID: 33188727) (Figure 4 in that paper). In additional, pre-miR-142 was shown to bear 0- to 3-nt 3'-overhang produced by Drosha (PMID: 24297910) (Figure 4 in that paper), a finding similar to our observation for pre-miR-217 in the present study.

Revision Figure 4 (Figure 4D in the main text). sRNA-Seq for ~20-nt long RNAs indicates consistency for Dicer cleavage-site selection at the 3p arms of pre-miR-217 variants, based on their spatial distance. The relative expression level and distance (in Å) are shown for each isoform (shown as a blunt-ended line).

The nucleotide substitutions on pre-miR217 variants are colored in red. The first nucleotides in the minor isoforms of hsa-miR-217-v2 are colored in green. Reactivity to NAI-N₃, assessed by *in vitro* SHAPE analysis, is shown as colored dots for select nucleotides. Red suggests high reactivity, while black indicates no reactivity.

We again thank the reviewer for the helpful guidance and his ongoing support of our manuscript.